**EMBO** *reports*

# Cellular remodeling and JAK inhibition promote zygotic gene expression in the *Ciona* germline

Naoyuki Ohta [1]✉ & Lionel Christiaen [1,2]✉

## Abstract

Transcription control is a major determinant of cell fate decisions in somatic tissues. By contrast, early germline fate specification in numerous vertebrate and invertebrate species relies extensively on RNA-level regulation, exerted on asymmetrically inherited maternal supplies, with little-to-no zygotic transcription. However delayed, a maternal-to-zygotic transition is nevertheless poised to complete the deployment of pre-gametic programs in the germline. Here, we focus on early germline specification in the tunicate *Ciona* to study zygotic genome activation. We first demonstrate that a peculiar cellular remodeling event excludes localized postplasmic *Pem-1* mRNA, which encodes the general inhibitor of transcription. Subsequently, zygotic transcription begins in *Pem-1*-negative primordial germ cells (PGCs), as revealed by histochemical detection of elongating RNA Polymerase II, and nascent *Mef2* transcripts. In addition, we uncover a provisional antagonism between JAK and MEK/BMPRI/GSK3 signaling, which controls the onset of zygotic gene expression, following cellular remodeling of PGCs. We propose a 2-step model for the onset of zygotic transcription in the *Ciona* germline and discuss the significance of germ plasm dislocation and remodeling in the context of developmental fate specification.

**Keywords** Tunicate; Primordial Germ Cells; Germline Specification; Transcription Control; Lobe Scission
**Subject Categories** Chromatin, Transcription & Genomics; Development; RNA Biology

## Introduction

During embryonic development, defined transitions in the composition of the cellular transcriptome and proteome govern successive cell fate decisions (Moris et al, 2016). Common features of fateful molecular transitions include (1) multilineage priming, whereby multipotent progenitors co-express determinants of distinct and mutually exclusive cellular identities (Nimmo et al, 2015; Razy-Krajka et al, 2014; Hu et al, 1997), (2) de novo gene expression, which adds to primed factors and completes fate-specific cellular programs

(Wang et al, 2019; Graf and Enver, 2009), and (3) cross-antagonisms, whereby competing cellular programs inhibit each other upon mutually exclusive fate choices (Wang et al, 2013). Transcriptional control exerts a dominant influence on these molecular transitions. Transcription regulators are thus widespread determinants of cell fate decisions, especially in somatic lineages (Levine and Tjian, 2003; Davidson and Levine, 2008; Levine and Davidson, 2005).

In mammals, early germ cell fate specification is also controlled by signal-mediated induction and transcriptional regulation (Ohinata et al, 2009; Tang et al, 2016; Mitsunaga and Shioda, 2018; Jostes and Schorle, 2018). By contrast, in other vertebrate species such as zebrafish and *Xenopus*, and in numerous invertebrate species, including the fly *Drosophila*, the nematode worm *C. elegans* and the ascidians *Halocynthia* and *Ciona*, early germ cell progenitors are transcriptionally silent (Nakamura and Seydoux, 2008; Kumano et al, 2011; Shirae-Kurabayashi et al, 2011). This transcriptional quiescence contributes to keeping germline progenitor cells from assuming somatic fates in response to inductive signals from surrounding cells in early embryos (Robert et al, 2015; Lebedeva et al, 2018). In these systems, the germline is set aside through unequal cleavages and asymmetric divisions, which segregates somatic lineages from primordial germ cells (PGCs), where transcription remains initially silent.

Early unequal cleavages are coupled with polarized distribution of maternal components including the germplasm, which carries global transcription inhibitors known in several invertebrate species, such as Pgc (polar granule component) in *Drosophila*, PIE-1 in *C. elegans* (Seydoux and Dunn, 1997; Batchelder et al, 1999; Blackwell, 2004; Seydoux and Braun, 2006; Hanyu-Nakamura et al, 2008), and Pem-1 in ascidians (Kumano et al, 2011; Shirae-Kurabayashi et al, 2011). Remarkably, although Pgc, PIE-1 and Pem-1 are unrelated proteins thought to have evolved independently in their corresponding phylogenetic lineages, they all inhibit transcription by blocking phosphorylation of Serine 2 in YSPTSPS-like heptapeptide repeats of the C-terminal domain of the RNA Polymerase II main subunit (RNAPII-CTD), which is necessary for transcriptional elongation (Lebedeva et al, 2018).

Consistent with the progressive segregation of transcriptional quiescence from the whole egg and early blastomeres to primordial germ cells, Pgc, Pie-1, and Pem-1 gene products are among the maternal components that constitute the germplasm and progressively segregate to PGCs (Mello et al, 1996; Nakamura et al, 1996; Yoshida et al, 1996). In ascidians, *Pem-1* belongs to a group of so-called postplasmic RNAs that are maternally deposited,

[1]Michael Sars Centre, University of Bergen, Bergen, Norway. [2]Center for Developmental Genetics, Department of Biology, New York University, New York, NY, USA.
✉E-mail: naoyuki.ohta@uib.no; lionel.christiaen@uib.no

accumulate to the vegetal-posterior end of the fertilized egg, and are inherited by the earliest germline progenitor cells, named B4.1, B5.2, B6.3, and B7.6, through subsequent unequal cleavages (Fig. 1D; Sasakura et al, 2000; Prodon et al, 2007). Consistent with the dominant effect of RNAPII inhibition by Pem-1, this lineage remains transcriptionally silent until a previously unknown stage.

Remarkably, when B7.6 blastomeres "divide" (As we show in this study, B7.6 cells do not actually divide, and the previously named B8.11 cell is actually a cellular fragment, which we herein call the lobe, by analogy with a phenomenon described in *C. elegans*.) during gastrulation, *Pem-1* mRNAs are asymmetrically inherited by only one of the "daughter cells", previously named B8.11, whereas B8.12, its *Pem-1* RNA-negative sibling, constitutes the bona fide primordial germ cell (PGC), the progeny of which later populates the somatic gonad in post-metamorphic juveniles (Shirae-Kurabayashi et al, 2006). Since *Pem-1* mRNAs are not inherited by PGCs, Pem-1 is likely dispensable for subsequent deployment of the germline-specific program in PGCs.

By contrast with *Pem-1* and several other postplasmic RNAs, mRNAs encoding the Vasa homolog Ddx4, a conserved RNA helicase involved in germ cell development in a broad range of species, are distributed into both *Pem-1*+ remnants and the PGCs (Shirae-Kurabayashi et al, 2006). Taken together, these observations suggest that maternal determinants of germline fate specification comprise both inhibitors of early somatic specification and primed regulators of the germline program, which segregate upon division of B7.6 blastomeres. We hypothesized that exclusion of *Pem-1* gene products licenses zygotic gene expression in PGCs, thus permitting the activation of de novo-expressed factors that complement the germline specification program.

More than 40 maternal RNAs have known postplasmic localization in the zygote and early ascidian embryo (Dehal et al, 2002; Yamada et al, 2005; Yamada, 2006). By contrast, there is limited-to-no information about zygotically expressed genes in the *Ciona* germline. Contrary to somatic lineages (Imai et al, 2006; Ohta and Satou, 2013; Satou and Imai, 2015; Oda-Ishii et al, 2016), general transcriptional quiescence has precluded traditional whole genome assays from informing early germline gene regulatory networks (GRNs).

Here, by monitoring the B7.6 lineage in *Ciona* embryos, we first observed that exclusion of *Pem-1* RNAs from the PGCs occurs, not by cell division as previously thought, but through a peculiar cell remodeling event that sheds postplasmic RNA-containing cytoplasm at the beginning of gastrulation. This cellular remodeling, and Pem-1 protein degradation (Shirae-Kurabayashi et al, 2011) are followed by initiation of transcription through the consecutive onsets of RNAPII activity and *Mef2* transcription, at neurula and tailbud stages. Finally, we uncovered a provisional antagonism between JAK and MEK/BMPR/GSK3 signaling that controls the timing of zygotic transcription initiation in the germline. Taken together, these results shed new light on an important transition in early germline development.

## Results and discussion

### Cellular remodeling excludes certain maternal postplasmic RNAs from primordial germ cells

In *Ciona* embryos, the B7.6 cells give birth to primordial germ cells (PGCs), which are thought to emerge after one more division, and

correspond to the B8.12 lineage, following the segregation of a subset of maternal postplasmic RNAs into their B8.11 sister cells (Takamura et al, 2002; Shirae-Kurabayashi et al, 2006). One of these RNAs, *Pem-1*, produces a nuclear protein that inhibits zygotic transcription in B7.6 cells, thus protecting the PGC lineage by preventing ectopic activation of somatic determinants (Kumano et al, 2011; Shirae-Kurabayashi et al, 2011). Therefore, we reasoned that zygotic genome activation might follow the exclusion of *Pem-1* RNA from the PGCs. To address this possibility, we first sought to identify candidate zygotically expressed genes, as well as reliable markers of B7.6 lineage cells. We leveraged the extensive in situ gene expression database ANISEED (Tassy et al, 2010; Dardaillon et al, 2020; Brozovic et al, 2018). Among genes encoding Postplasmic/PEM RNAs maternally expressed and localized in the B7.6 lineage, we used *Pem-1* (KH.C1.755; Yamada et al, 2005) as a B8.11 marker, and *Ms4a15/2* (KH.C2.4, aka *Pem-7*; (Nishikata et al, 2001; Yamada, 2006)) as a dual B8.11 /B8.12 marker. Whole-mount fluorescent in situ hybridization (FISH) assays confirmed the expected localization of *Pem-1* and *Ms4a15/2* mRNAs in small B8.11 and large B8.12 cells at the mid tailbud stage (stage 21 (Hotta et al, 2007); Fig. 1A,B). To our surprise, we did not observe any DAPI-positive nucleus in *Pem-1* + B8.11 "cells" in tailbud embryos (Fig. 1C–C'), a pattern visible, but seemingly unnoticed, in a previous publication (Fig. 6O in Shirae-Kurabayashi et al, 2006).

This observation prompted us to re-evaluate whether B7.6 cells undergo bona fide cell divisions, or cellular remodeling events akin to lobe formation and scission in the primordial germ cells of *C. elegans* (Abdu et al, 2016). To this aim, we used cell-specific DiI labeling to monitor B7.6 cell shape changes from the gastrula stage onward (Fig. 1D–H'; (Satou et al, 2004; Shirae-Kurabayashi et al, 2006)). We observed DiI+ cell fragments separate from B7.6 cells (Fig. 1F–G'), and lacking DAPI+ nuclei as early as the early neurula stage (Fig. 1H–H'). Live imaging following B7.6 cells staining with the BioTracker MemBright dye supported the notion that lobes are generated from B7.6 cells (Fig. 1I,J; Movies EV1 and EV2).

To test whether these B7.6-derived cell fragments correspond to the entities previously recognized as B8.11 cells, we performed fluorescent in situ hybridization using the *Pem-1* probe on DiI-labeled neurula stage embryos. This experiment showed colocalization of *Pem-1* RNA with B7.6-derived cell fragments in neurula stage embryos (Fig. 1K–L"). Even though postplasmic RNAs localize the posterior end of the fertilized eggs and early embryos, lobes tended to segregate from the anterior end of B7.6 cells. To clarify this potential conundrum, we inspected embryos collected in time series encompassing lobe formation and scission in embryos. As expected, *Ms4a15/2* mRNAs localized the posterior end of B6.3 cells at 32-cell stage, and the mRNAs were still inherited to the posterior end of B7.6 cells at the beginning of 64-cell stage. During gastrulation, however, the FISH signal became visible anteriorly, suggesting a relocalization of postplasmic mRNAs, reflecting changes in B7.6 cell polarity prior to lobe formation (Appendix Fig. S1A,B). Supporting a role for active actin dynamics in lobe formation and *Pem-1* mRNA localization in B7.6 cells, we sprayed Cytochalasin D specifically onto B7.6 cells, which blocked the lobe formation and disrupted *Pem-1* RNA concentration, causing it to become distributed throughout the whole B7.6 cells (Fig. 1M–Q).

To further probe whether postplasmic RNA exclusion from PGCs occurs via cell division or remodeling, we monitored

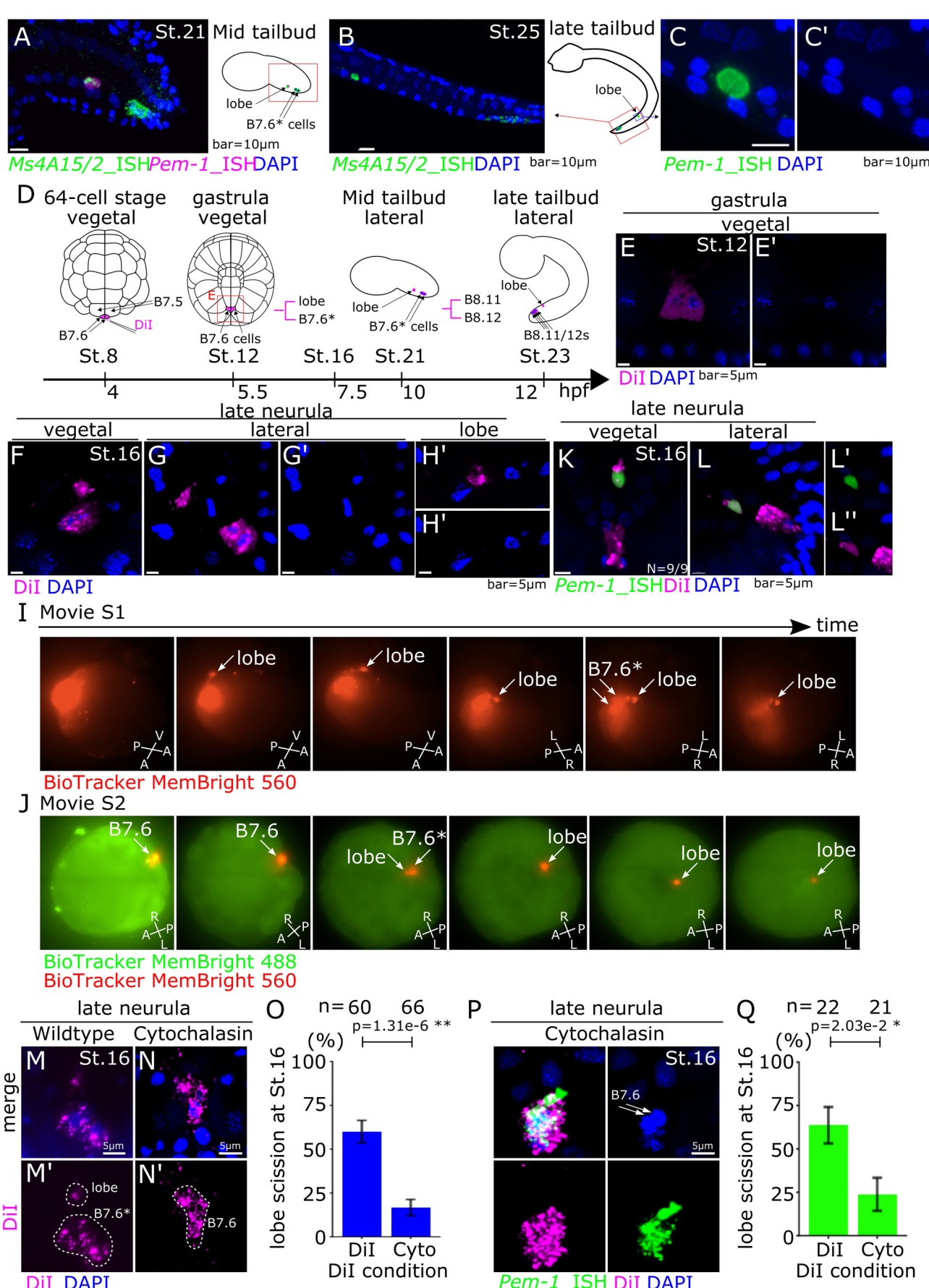

**Figure 1. B7.6 cells form a lobe including *Pem-1* mRNA without nuclei.**

(A) Expression of PGC marker genes, *Pem-1* and *MS4a15/2* detected by ISH probes corresponding to exons at the mid tailbud stage, St. 21. (B) Expression of *MS4a15/2* at the late tailbud stage, St. 25. (C, C') Expression of *Pem-1* in lobe at the late tailbud stage, St. 25. C' shows the blue channel of (C). (D) Schematic diagram of *Ciona* development from 64-cell stage, St. 8 to late tailbud stage, St. 23. Magenta shows DiI label. (E–H') Cell membranes of B7.6 cells were stained by DiI at the 64-cell stage, and images taken at the gastrula (E, E') and the neurula stages (F–H'). (E', G', H') Blue channels of (E, G, H), respectively. (I, J) B7.6 cells were stained by spraying with BioTracker Membright 560 Dye (I), and the whole cell membrane was stained with BioTracker Membright 488 (J). Snapshots of Movies EV1 and EV2 are shown in time from left to right. (K, L) B7.6 cells were traced by DiI, and lobe was detected by ISH with a *Pem-1* probe corresponding to exons. (L') Green and blue channels of (L). (L") Magenta and blue channels of (L). (M–Q) DiI and Cytochalasin D were specifically sprayed at B7.6 cells, and lobe scission was observed and quantified as proportion of successful lobe scission based on the DiI staining (M–Q), and the segregation of the Pem-1 mRNA was observed (P, Q). Data information: (O, Q) n means the number of embryos. Error bars indicate standard error. *P* value was calculated by z-test. *P* > 0.05; N.S, 0.05>*P* > 0.01; *, 0.01>*P*; **. (A–C') Scale bars are 10 μm. (E–H', K–L', M, N', P) scale bars are 5 μm. Source data are available online for this figure.

phosphohistone 3 (pH3) alongside Ms4a15/2 mRNAs to jointly detect mitotic activity and postplasmic RNA localization in B7.6 cells through gastrulation (Appendix Fig. S1C,D). Clear pH3 signal in dividing B6.3 and some somatic blastomeres confirmed our ability to detect mitotic cells. However, in contrast to a previous report (Shirae-Kurabayashi et al, 2006), we observed no pH3 signal in B7.6 nuclei throughout postplasmic RNA repolarization and in the lead up to lobe formation and scission. These observations indicate that B7.6 cells repolarize and undergo a cell remodeling event toward the end of gastrulation, which results in the shedding of cytoplasm containing maternal postplasmic RNA including *Pem-1*, into a cell fragment that we refer to as "lobe", by analogy with the PGC remodeling process described in *C. elegans* (Mainpal et al, 2015; Abdu et al, 2016; Maniscalco et al, 2020; McIntyre and Nance, 2020). On the other hand, remodeled B7.6 cells, which we propose to call B7.6*, are the bona fide primordial germ cells in *Ciona*.

## PGC lobes remain associated with endoderm progenitors

In various animal species, PGCs associate with endodermal progenitors (Ying et al, 2002; Pilato et al, 2013). In *C. elegans* for instance, intestinal precursors actively phagocytose the germline lobes (Abdu et al, 2016). In ascidians, B7.6 cells abut the posterior-most endodermal progenitors, and the PGCs remain associated with the intestinal anlage, known as endodermal strand, in the larval tail (Nishida and Satoh, 1983; Takamura et al, 2002; Kawai et al, 2015). We thus explored a possible involvement of the endoderm in PGC remodeling. We combined fluorescent immunohistochemical (IHC) staining and whole mount in situ hybridization to jointly detect the endoderm reporter *Nkx2-1 > hCD4::mCherry* (*Nkx2-1* enhancer driving human cluster of differentiation 4 conjugated to monomeric Cherry fluorescent protein) (Ristoratore et al, 1999; Gline et al, 2015) and the B7.6 lineage marker *Ms4a15/2* in neurula stage embryos. These assays indicated that, while the B7.6 cell bodies remained adjacent to, but outside, the endoderm, the lobe appeared wedged in between endoderm progenitor cells (Appendix Fig. S2A,B"). Likewise, joint visualization of endodermal cell membranes, labeled by *Nkx2-1 > hCD4::GFP*, and DiI-labeled B7.6 cells revealed the proximity between detached lobes and endodermal progenitors (Appendix Fig. S2C–E). However, by contrast with *C. elegans*, we did not obtain clear evidence that endoderm cells engulf PGC lobes in *Ciona*. Taken together, our data indicate that B7.6 PGC progenitor cells undergo actin-driven cellular remodeling events that produce cellular fragments, the lobes, which contain maternally deposited

postplasmic mRNAs, and remain in close proximity to endodermal intestine progenitor cells.

## Cellular remodeling precedes the onset of zygotic transcription in PGCs

In ascidians, the maternal postplasmic mRNA *Pem-1* encodes a nuclear protein that inhibits zygotic transcription (Kumano et al, 2011; Shirae-Kurabayashi et al, 2011). Pem-1 acts by blocking the phosphorylation of Serine 2 in heptapeptide repeats at the RNA polymerase II C-terminal domain (RNAPII-CTD) (Kumano et al, 2011). We thus reasoned that, by removing *Pem-1* mRNA from B7.6 cells, cellular remodeling may contribute to activating zygotic gene expression in the germline. Indeed, previous immunostaining assays indicated that Pem-1 protein localizes to both nucleus and lobe during B7.6 cell remodeling, but became undetectable in PGCs by the tailbud stage (Shirae-Kurabayashi et al, 2011). To test whether the removal of Pem-1 licensed zygotic transcription in remodeled PGCs, we labeled B7.6 cells with DiI, and fixed embryos between the early gastrula and tailbud stages (6–12 h post fertilization (hpf) at 18 °C; St. 13, 16, 21, and 23) for immunostaining with an anti-RNAPII-CTD-pSer2 antibody (Fig. 2A–D). Consistent with previous reports (Shirae-Kurabayashi et al, 2011), we did not detect RNAPII-CTD-Ser2 phosphorylation in B7.6 cells at 6 hpf (St. 13). By contrast, the majority of mid-tailbud stage embryos displayed conspicuous RNAPII-CTD-Ser2 phosphorylation in PGC nuclei (2 nuclei at 10 hpf, and 4 nuclei at 12 hpf after one cell division of B7.6* cells), from 10 hpf (St. 21) onward. These results indicated that transcription elongation by RNA polymerase II is active in the PGCs of mid-tailbud embryos, which follows the exclusion of *Pem-1* mRNAs by B7.6 cell remodeling and presumably Pem-1 protein degradation in PGCs (Shirae-Kurabayashi et al, 2011).

## *Mef2* is zygotically transcribed in the PGCs

Having established that the PGCs of mid-tailbud embryos are transcriptionally active, we sought to identify genes expressed zygotically in B7.6* cells. We mined the gene expression database ANISEED (Tassy et al, 2010; Dardaillon et al, 2020; Brozovic et al, 2018), and identified candidate transcription factor-coding genes possibly upregulated in B7.6* cells after the exclusion of postplasmic RNAs by lobe scission. Here, we focus on a transcription factor coding gene, *Myocyte elongation factor 2* (*Mef2*; KH.S455.6). *Mef2* maternal mRNAs were detected

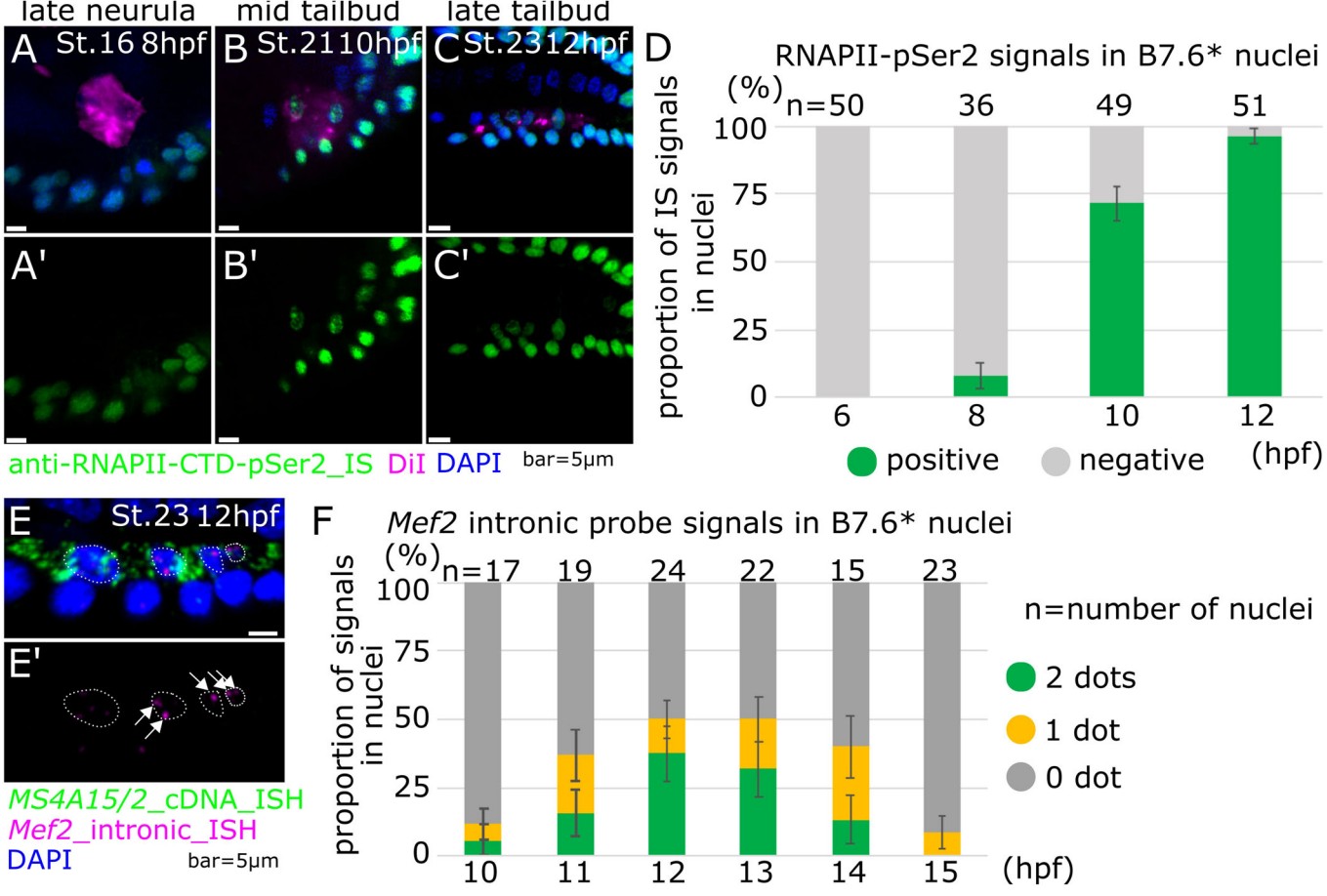

**Figure 2. RNA polymerase II Ser2 phosphorylation and *Mef2* nascent expression in B7.6* cells.**

(A–C) Immunostaining for anti-RNAPII-CTD-pSer2 antibody was done at the late neurula; 8 hpf (A), the mid tailbud; 10 hpf (B) and the late tailbud; 12 hpf (C) stages. (D) Proportion of RNAPII-CTD-pSer2 positive B7.6* cells. (E–E') Nascent expression of *Mef2* gene was detected by *Mef2* intronic probes in 12 hpf embryos at 18 °C. White arrows indicate the dotted signals of nascent *Mef2* expression in nuclei of B7.6* cells. White dotted circles show nuclei of B7.6* cells. (F) *Mef2* intronic probe signals in nuclei in B7.6* cells. The proportion of signal positive nuclei was shown in y axis. Data information: (D, F) Error bars indicate standard error. *n* means the number of nuclei. (A–C, E) Scale bars are 5 μm. Source data are available online for this figure.

ubiquitously in the whole early embryo and intense ISH signals were observed in the muscle, endoderm and epidermis (Imai, 2004). This abundance of maternal mRNA prevented us from identifying zygotic products with cDNA-derived probes. To specifically monitor zygotic *Mef2* expression, we synthesized *Mef2*-specific intronic probes, which detect nascent transcripts, as shown for *Gata4/5/6*, *Tbx1/10* and *Ebf* in the cardiopharyngeal lineage (Wang et al, 2013; Razy-Krajka et al, 2018). To unequivocally identify PGCs, we performed double FISH assays with *Mef2* intronic probe and the B7.6 lineage marker *Ms4a15/2* (Fig. 2E). Mid- to late-tailbud embryos raised at 18 °C, and collected in a time series between 10 and 15 hpf, St. 21–24, showed conspicuous nuclear dots of nascent transcripts, indicative of zygotic *Mef2* expression in B7.6*, as well as epidermal and muscle cells (Fig. 2E–E'). Because transcription was shown to occur in bursts of RNA polymerase activity in other systems (Gregor et al, 2014; Tkačik and Gregor, 2021), and our assay provided snapshots of dynamic nuclear states, we reasoned that *Mef2* loci in B7.6* cells might be transcriptionally active and yet show 0, 1, or 2 dots (assuming that even after DNA replication, sister chromatids

remain too close to distinguish by standard confocal microscopy). We thus counted *Mef2*+ dots per B7.6* nucleus at successive stages, to obtain a semi-quantitative view of transcriptional activity at the *Mef2* locus in developing PGCs. This analysis indicated that zygotic *Mef2* expression peaked at the late tailbud I stage (st. 23, 12 hpf at 18 °C), from an onset around 10 hpf, and was markedly downregulated by 15 hpf (Fig. 2F).

At its peak, we detected 1 or 2 fluorescent dots in 50% of the nuclei, and thus sought to further verify that these correspond to active *Mef2* transcription. We treated embryos with known chemical inhibitors of transcription and assayed zygotic *Mef2* expression with our intronic probe. Actinomycin D, an established transcription inhibitor that binds "melted" DNA at the pre-initiation complex and prevents RNA elongation (Sobell, 1985), was previously used on ascidian embryos (Shirae-Kurabayashi et al, 2006; Miyaoku et al, 2018), and caused a modest but significant down-regulation of *Mef2* expression in B7.6* cells (Appendix Fig. S3). On the other hand, the CDK9 inhibitor Flavopiridol, which also blocks transcriptional elongation, but by preventing RNAPII-CTD-Ser2 phosphorylation (Bensaude, 2011), nearly

  

eliminated *Mef2*+ nuclear dots (Appendix Fig. S3). In either case, chemical inhibitor treatments supported the interpretation that intronic probe-positive nuclear dots represent nascent transcripts, which is indicative of zygotic *Mef2* expression in the PGCs during the tailbud developmental period. Notably, the temporal profile of zygotic *Mef2* expression showed an onset that followed both B7.6 cell remodeling and the global activation of RNA Polymerase II. This is consistent with a causal chain of events whereby postplasmic RNA exclusion through cellular remodeling helped relieve the Pem-1 break on RNAPII activity, thus permitting subsequent zygotic gene expression in the PGCs.

## JAK signaling delays the onset of zygotic *Mef2* expression in B7.6* cells

Having established that cellular remodeling precedes global RNAPII licensing and the onset of *Mef2* transcription in B7.6* cells, we sought to identify regulators of zygotic *Mef2* expression in the PGCs. We reasoned that signaling inputs from surrounding cells probably contribute to initiating zygotic gene expression in B7.6* cells, and their possible roles are readily testable by pharmacological inhibition. Treatments with U0126 (10 μM; 10–12 hpf; St. 21–23), Dorsomorphin (10 μM; 10–12 hpf) and 1-Azakenpaullone (10 μM; 10–12 hpf), which inhibit MEK1/2, BMPRI and GSK3, respectively, significantly reduced the proportions of nuclei with detectable nascent *Mef2* transcripts in *Ms4a15/2*+ PGCs (Appendix Fig. S4). However, neither of these inhibitors completely abolished zygotic *Mef2* expression. We thus tested whether these signaling pathways act additively, by combining the three inhibitors in a condition referred to as "3i" (10 μM each; 10–12 hpf), which indeed caused the most marked decrease in zygotic *Mef2* expression in B7.6* cells (Fig. 3A,B,E). This suggested that MEK, BMPR and GSK3, whose possible sources are in the vicinity of PGCs at the tail tip and ventral side of the tail (Imai, 2004; Waki et al, 2015; Feinberg et al, 2019; Harder et al, 2019), act at least partially additively to promote zygotic *Mef2* expression in the PGCs.

By contrast with the 3i treatment, the established JAK inhibitor Ruxolitinib (10 μM; 10–12 hpf) significantly increased the fraction of PGCs with *Mef2*+ nuclei, in addition to inhibiting B7.6* cell division and tail elongation (Fig. 3C,E,F). To test a possible hierarchy between the signaling pathways that regulate *Mef2* transcription in the PGCs, we combined the 3i and Ruxolitinib treatments (Fig. 3D,E). This combination also increased the proportions of the *Mef2*+ nuclei, to levels similar to those obtained with Ruxolitinib alone. Taken together, these results suggest that JAK signaling inhibits zygotic *Mef2* expression in B7.6* cells, while MEK, BMPR, and GSK3 act additively to promote *Mef2* transcription, antagonizing JAK activity.

To gain further insights into the biological significance of inhibitor treatments and better characterize the context in which JAK signaling regulates zygotic *Mef2* expression in the PGCs, we sought to assay endogenous JAK activity in *Ciona* embryos. Previous genome-wide surveys identified two *Jak* ortholog (*Jak-a*: KH.C1.555 and *Jak-b*: KH.C8.409) in the *Ciona* genome (Hino et al, 2003; Tokuoka et al, 2022). The *Jak-a* RNAs are maternally deposited in unfertilized eggs as detected by ISH, ESTs and bulk RNA-seq (Appendix Table S2) (Satou et al, 2005; Dardaillon et al, 2020). *Ciona* Jak-a displays one region surrounding a potential

active phosphorylated tyrosine residue that is conserved with human JAK2 (Appendix Fig. S5A). We thus tested an anti-phospho-human JAK2 (Y931) polyclonal antibody to detect endogenous JAK activity in tailbud embryos, and observed conspicuous signal in tail tip and dorsal midline cells (Fig. 3G,I). Treatment with the JAK inhibitor Ruxolitinib abolished the signal, which is consistent with the specificity of both the drug treatment and antibody staining (Appendix Fig. S5B–G). JAK activity was most conspicuous at the tail tip of St. 21 embryos (10 hpf at 18 °C), and markedly reduced by St. 23 (12 hpf at 18 °C), which coincides with peak transcriptional activity of the *Mef2* gene in the germline at St. 23 (Fig. 2F). While this pattern is consistent with concomitant down-regulation of JAK signaling and activation of *Mef2* expression, as well as the effect of Ruxolitinib treatments, the latter do not distinguish between cell-autonomous and non cell-autonomous effects in B7.6* cells. To evaluate the possibility that JAK acts cell-autonomously upstream of *Mef2* to repress its transcription in the PGC lineage, we DiI-labeled B7.6 cells and assayed JAK activity at two developmental stages corresponding to the onset and peak of zygotic *Mef2* expression (Fig. 3G–K). This experiment revealed JAK activity in B7.6* cells at St. 21 (10 hpf at 18 °C), but reduced signaling by St. 23 (12 hpf at 18 °C), which is consistent with a cell-autonomous inhibitory effect on early *Mef2* transcription, although it does not formally rule out non-cell autonomous influence from the distal tail epidermis.

Of note, the localization of maternal *Jak-a* mRNAs in early embryos, opens the possibility that JAK signaling is active in early B7.6 blastomeres. We tested endogenous JAK activity at the early developmental stage, and observed signals at the vegetal side of the cell membrane on B7.6 cells and endodermal cells (Appendix Fig. S6A). JAK activity initiates during the 110-cell to early gastrulation stage (Appendix Fig. S6B,C). Treatment with a translation-inhibitor antisense morpholino oligonucleotide conjugated vivo (vivo-MO) targeting the 5' untranslated region (5' UTR) of *Jak-a* mRNA abolished the anti-pJAK2 signal, indicating that maternal *Jak-a* mediates JAK signaling, and further supporting the notion that both the vivo-MO treatment and antibody staining are specific tools to study JAK signaling in *Ciona* (Appendix Fig. S6C,D). In summary, these data indicate that maternal *Jak-a* mRNA fosters JAK signaling in the germline, from early gastrulation and until the tailbud stage.

Leveraging anti-pJAK immunostaining, we tested whether MEK, BMPR and GSK3 signaling contribute to inhibiting JAK phosphorylation by stage 23 by staining embryos treated with the 3i cocktail of inhibitors (Appendix Fig. S7). We observed no conspicuous difference between 3i and DMSO control treatments, suggesting that Jak dephosphorylation occurs independently of MEK, BMPRI, GSK3 signaling inputs (Appendix Fig. S7).

Taken together, our observations are consistent with the possibility that JAK signaling plays an early inhibitory role in delaying the peak of *Mef2* transcription in the PGCs. Indeed, RNA polymerase II is licensed to initiate transcription as early as 8 hpf (Fig. 2D; St. 16), while zygotic *Mef2* expression does not start before 10 hpf, and only peaks at 12 hpf (Fig. 2F). To test if JAK signaling inhibits early *Mef2* expression, we thus treated embryos with Ruxolitinib from 8 hpf onward (St. 16), and assays zygotic *Mef2* expression at 10 hpf (Fig. 3L–O). In control DMSO-treated embryos, at most 25% of the B7.6 lineage nuclei showed nascent *Mef2* transcripts. By contrast, we detected active *Mef2* transcription

  

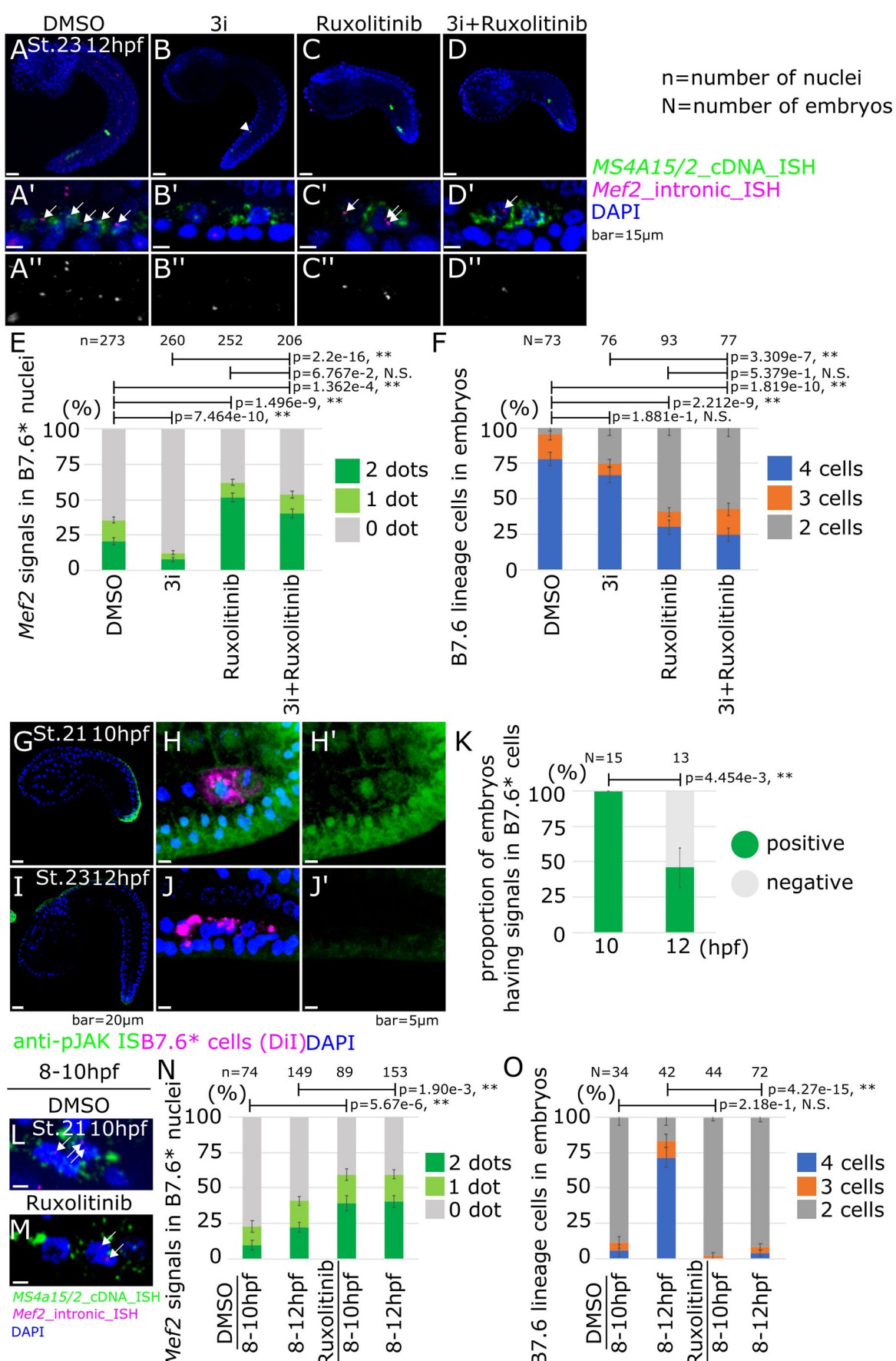

◄ **Figure 3.** *Mef2* nascent expression in inhibitor-treated embryos.

(A–D) ISH was done with *Mef2* intronic probes under each pharmacological inhibitor treatment; DMSO (A), 3i (B), Ruxolitinib (C) and 3i+Ruxolitinib (D). White arrows show the dotted signals of nascent *Mef2* expression. White arrowhead shows lobe in (B). (A′–D′) Magnification images of (A–D). (A″–D″) Black and white images of the magenta channel of (A′–D′), respectively. (E) Proportion of B7.6* cells with *Mef2* signals in nuclei (y axis) under each pharmacological inhibitor treatment (x axis). (F) Proportion of B7.6* lineage cells in embryos (y axis) under each pharmacological inhibitor treatment. (x axis). (G–J) Immunostaining with anti-JAK2 antibody was done at the mid tailbud; 10 hpf (G–H′) and the late tailbud; 12 hpf (I–J′) stages. Image G is the same as the image on S5B. (K) Proportion of embryos with JAK2-positive B7.6* cells (y axis). (L, M) ISH was done with *Mef2* intronic probes under each pharmacological inhibitor treatment; DMSO (L) and Ruxolitinib (M) in 10 hpf embryos and quantified in (N). White arrows show the dotted signals of nascent *Mef2* expression. (N) Proportion of *Mef2*-positive B7.6* nuclei showing either 1 or 2 dots (y axis) under each condition. (O) Proportion of cell numbers of B7.6* cells in embryos (y axis) under each condition. Data information: (E, F, K, N, O) Error bars indicate standard error. P value was calculated by z-test. P > 0.05; N.S, 0.05>P > 0.01; *, 0.01>P; **. Scale bars are 15 μm on (A–D), 20 μm on (G, I) and 5 μm on (H, J, L, M). Source data are available online for this figure.

---

in ~50% of the B7.6 lineage nuclei following treatment with the JAK inhibitor between 8 and 10 hpf (St. 16–21; Fig. 3L–O). Moreover, 8 to 12 hpf (St. 16-23) treatments resulted in similar ~50% of *Mef2*^intron^ + B7.6 lineage nuclei, albeit with a lower fold-increase, as the DMSO baseline was at 41%, suggesting that *Mef2* transcription peaks earlier following early JAK inhibition. Of note, control B7.6* cells had not divided in the 8–10 hpf time window (Fig. 3O). These early treatments thus indicated that chemical JAK inhibition promotes *Mef2* transcription independently of its effects on cell division. Taken together, these results support the notion that early, possibly cell-autonomous, JAK activity delays the peak, and probably the onset of zygotic *Mef2* expression in the PGCs.

## MEK, BMPR, GSK3, and JAK signaling regulate *Mef2* transcription independently of RNA polymerase II CTD phosphorylation

Finally, since global transcription licensing shortly precedes the onset of zygotic *Mef2* expression, we tested whether the kinases that influence the production of nascent *Mef2* transcripts also impact RNAPII-CTD phosphorylation. To this aim, we repeated the above chemical inhibitor treatments targeting MEK, BMPRI, GSK3 and JAK, and assayed RNAPII-CTD-Ser2 phosphorylation at 8 hpf, St. 16 and 12 hpf, St. 23. In short, neither of these inhibitor treatments appeared to alter RNA polymerase II activity (Fig. 4A–E; Appendix Fig. S8), suggesting that the kinases regulate zygotic *Mef2* expression independently of global RNAPII activation (Fig. 4G). Animal species endowed with "preformed" primordial germ cells illustrate August Weismann's famous dichotomy between mortal somatic vs. immortal germline lineages, and the mechanisms underlying this divergence in early embryos have garnered much attention (Kutschera and Niklas, 2004; Dröscher, 2014; Weismann, 1892). Despite the de facto immortality and totipotency of their lineage, illustrated by their ability to reconstitute an entire organism upon fertilization and embryogenesis, PGCs eventually assume an identity and differentiate into highly specialized egg and sperm cells, the gametes. As cells, PGCs are thus bound to the same developmental constraints as somatic cells: to choose an identity, which comprises avoiding others, and differentiate accordingly, with the notable distinction that germ cells may postpone terminal differentiation until sexual maturity (Kang and Han, 2011; Hayashi and Saitou, 2014; Bhartiya et al, 2017).

This opportunity to pause differentiation may relate to the observed delay in starting zygotic gene expression in the germline, which is a widespread mechanism used to prevent somatic fate specification (Strome and Updike, 2015). Indeed, the unrelated

proteins Pgc, PIE1 and Pem-1 globally prevent zygotic gene expression in the germline by inhibiting phosphorylation of Serine 2 in heptapeptide repeats of the C-terminal domain of RNA Polymerase II, in as diverse animals as Drosophila, *Caenorhabditis elegans* and ascidians, respectively (Seydoux and Dunn, 1997; Batchelder et al, 1999; Blackwell, 2004; Seydoux and Braun, 2006; Hanyu-Nakamura et al, 2008).

This early global inhibition of transcription in PGCs imposes an exclusive reliance on maternal products deposited in the egg during oogenesis (Lehmann, 2012; Trcek and Lehmann, 2019). Among those, germline-specific components are typically found in RNA-rich germ granules, germplasm or nuage, and dispatched into PGCs through polarized RNA and protein localization, and asymmetric inheritance following unequal cleavages in early embryos (Sardet et al, 2007; Little et al, 2015; Trcek and Lehmann, 2019). In early ascidian embryos, *Pem-1* maternal mRNAs are localized and associated with the cortical centrosome attracting body (CAB), segregate asymmetrically at every unequal cleavage, and are found exclusively in B7.6 blastomeres by the 64-cell stage (Fig. 4F) (Sardet et al, 2007). While the Pem-1 protein keeps B7.6 cells, and their progenitors, transcriptionally quiescent, it does not have a direct role in specifying the germline identity, which is another function that must be partially carried by the germplasm, in the absence of zygotic gene expression (Strome and Updike, 2015; Nakamura and Seydoux, 2008). In *Ciona*, as in numerous other species, the conserved RNA helicase Vasa/Ddx4 is thought to act as a PGC determinant (Takamura et al, 2002). Vasa/Ddx4 mRNAs and proteins are segregated mainly to B7.6* cells/PGCs, as is the case for P granules in *C. elegans* (Paix et al, 2009; Abdu et al, 2016). The germplasm must thus fulfill two essentials, but partially antagonistic functions for germline specification: (1) inhibiting the alternative somatic fate specification through transcriptional silencing, and (2) promoting the acquisition of germline-specific molecular features, including transcriptional activation of a zygotic germline program. We must thus anticipate that germplasm remodeling and elimination of anti-somatic and pro-germline activities is essential for germline development. Consistent with this assertion, germplasm remodeling has been observed, and segregation of its components is important for germline development in flies (Little et al, 2015) and zebrafish (D'Orazio et al, 2021).

Here, we presented data that corroborate previous observations about the dislocation of the germplasm and differential segregation of its components in *Ciona*, and showed that this results from a peculiar cellular remodeling event rather than unequal cell cleavage, as previously thought based on phospho-histone 3 staining in B7.6 (Shirae-Kurabayashi et al, 2006). The vesicles had thus been referred to as B8.11 cells, but did not show DAPI+ staining or evidence of

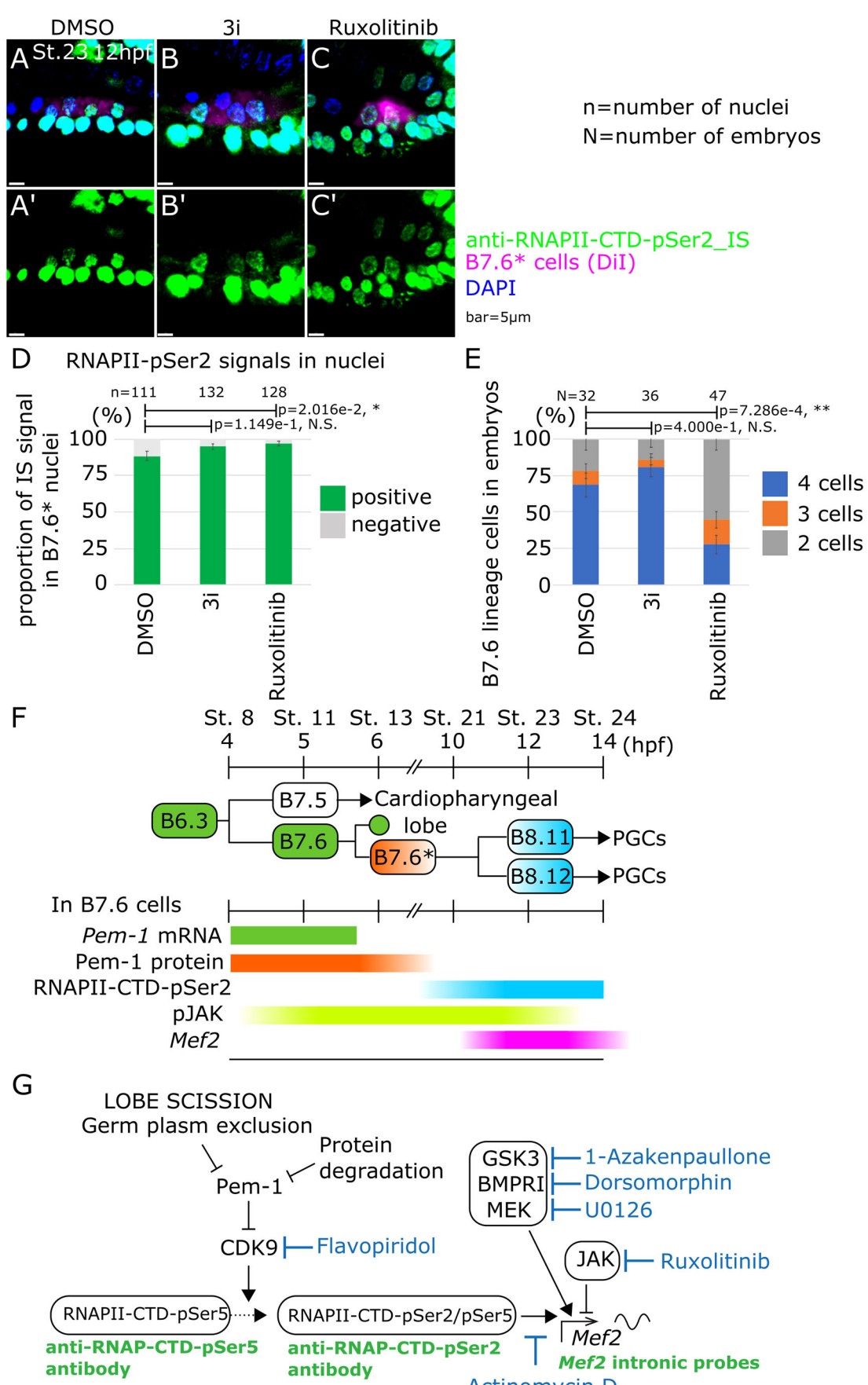

Figure 4. **RNA polymerase II Ser2 phosphorylation in inhibitor-treated embryos.**

(A–C) Immunostaining was done with anti-RNAPII-CTD-pSer2 antibody under each pharmacological inhibitor treatment at 12 hpf. (D) Proportion of B7.6* cells with a positive signal of immunostaining with anti-RNAPII-CTD-pSer2 antibody in the nucleus (*y* axis) under each pharmacological inhibitor treatment. (E) Proportion of B7.6* cell numbers in embryos (*y* axis) under each pharmacological inhibitor treatment. (F) A schematic image for activity or presence of each factor in B7.6 and B7.6* cells based on this study and a previous report (Shirae-Kurabayashi et al, 2011). (G) Working scheme of onset of zygotic transcription in B7.6* cells. Data information: (D, E) Error bars indicate standard error. *P* value was calculated by z-test. *P* > 0.05; N.S, 0.05 > *P* > 0.01; *, 0.01 > *P*; **. (A–C″) Scale bar is 5 μm. Source data are available online for this figure.

nucleokinesis, and are not actual cells. We propose to call them "lobes", after the analogous entity observed in *C. elegans*. Moreover, *Pem-1* mRNAs are among the postplasmic RNAs that segregate specifically to the lobe (Yamada, 2006; Paix et al, 2009), presumably contributing to the elimination of Pem-1 proteins from the PGCs by the tailbud stage (Shirae-Kurabayashi et al, 2011). Similar patterns of postplasmic RNA segregations were observed in other tunicates, like the Appendicularian, *Oikopleura dioica* (Olsen et al, 2018). Since Pem-1 globally blocks transcription, we propose that lobe scission is an important step to physically remove postplasmic RNAs from the PGCs for preventing synthesis of additional Pem-1 protein there, and permit further germline specification, in part through zygotic gene expression. In another ascidian species, *Halocynthia roretzi*, removal of *Pem-1* gene products appears to be necessary and sufficient for zygotic expression of ATP/ADP translocase gene in the germline; however, the cellular and molecular mechanisms likely differ as a sizeable portion of the *HrPEM* RNAs are retained in B7.6* PGCs (Miyaoku et al, 2018).

Germline lobe formation and scission exhibit intriguing parallels and differences between *Ciona* and *C. elegans*. One of the hallmarks of lobe formation and scission in *C. elegans* is the polarized distribution and removal of most mitochondria from the cell bodies of PGCs (Abdu et al, 2016; Schwartz et al, 2022). By contrast, in *Ciona* embryos, most mitochondria are depleted from B7.6 cells during the asymmetric division of their mother B6.3 cell, which also produces the B7.5 cardiac progenitors that inherit most mitochondria (a.k.a. myoplasm in ascidians)(Chenevert et al, 2013). In *Ciona*, mitochondria clearance from the germline is thus decoupled from lobe formation.

Another intriguing parallel between germline lobe formation and scission in *C. elegans* and *Ciona* regards its association with endodermal progenitors. Primordial germ cells exhibit an evolutionary conserved association with the endoderm in a variety of species (Anderson et al, 2000; Kanamori et al, 2019), including *Ciona* (Karaiskou et al, 2015; Kawai et al, 2015; Krasovec et al, 2019). The physical proximity between lobes and endoderm cells could thus be shared between *Ciona* and *C. elegans* as a byproduct of an ancient association between PGCs and endodermal progenitors. As lobe formation in *C. elegans* is inhibited by loss of dynamin and Rac function in the endoderm (Abdu et al, 2016; Maniscalco et al, 2020), germline lobes processing in *Ciona* could also involve the endoderm. Further perturbation of endodermal functions would be required to test this hypothesis, and may reveal additional cellular differences in the relationships between the endoderm and the germline in *Ciona* and *C. elegans*.

Finally, much attention has been devoted to the molecular mechanisms underlying germ plasm structure and function during germline formation (Seydoux and Braun, 2006; Lehmann, 2012; Trcek and Lehmann, 2019). By contrast, less is known about the mechanisms involving cell–cell signaling and zygotic gene expression in early germline specification, with the notable exception of PGC specification in mammals, which is governed by FGF, BMP and Wnt signaling pathways, as well as leukemia inhibitory factor (LIF)-controlled signaling (Lawson et al, 1999; Ohinata et al, 2009; Yu et al, 2020). Here, having identified one zygotically expressed gene in *Ciona* PGCs, we began to disentangle the mechanisms governing early genome activation in the germline. Specifically, chemical inhibitor treatments suggest that MEK, BMPRI and GSK3 contribute to activating *Mef2* transcription in the PGCs, providing JAK activity is inhibited. Our observations indicate that early JAK signaling acts independently of Pem-1-mediated inhibition of RNAPII to inhibit *Mef2* transcription, and after lobe scission to delay the onset of *Mef2* expression. Whereas further studies will be required to dissect cell autonomous or non cell autonomous effects in B7.6* cells, and to determine the source of signals and the genome-wide extent of zygotic gene expression in the *Ciona* germline, these results evoke a parallel between the signaling circuits that control germline specification in *Ciona*, and those that maintain stemness in mammalian pluripotent stem cells, which also rely on active JAK signaling and the concurrent inhibition of MEK and GSK3 (Ying et al, 2008; Hanna et al, 2010; Gafni et al, 2013; Ware et al, 2014; Takashima et al, 2015).

# Methods

## Animals

Wild-type animals of *Ciona intestinalis* type A were collected by M-Rep (Marine Research and Educational Products'), in San Diego, CA, and *Ciona intestinalis* type B were collected in Bergen, Norway. While *Ciona intestinalis* type A and type B have recently been considered as different species (Brunetti et al, 2015), they can cross each other to establish hybrid animals, who can be raised to at least F3 generation, suggesting their conserved developmental process (Suzuki et al, 2005; Sato et al, 2014; Malfant et al, 2017; Ohta et al, 2020). In this study, we used both *Ciona* species in mixture. Especially, data in Appendix Figs. S1 and S6 and Movies EV1 and EV2 were taken in *Ciona intestinalis* type B, and the others were taken in *Ciona intestinalis* type A. Eggs and sperm were surgically collected from mature adults. Chorion of fertilized eggs were removed by Sodium thioglycolate and Proteinase as previously described (Christiaen et al, 2009a). Dechorionated eggs were cultured on agarose-coated Petri dishes in TAPS-buffered artificial seawater (ASW; Bio actif sea salt, Tropic Marin).

## DiI cell tracing and blocking lobe scission

CellTracker CM-DiI Dye (Thermo Fisher Scientific) was dissolved in DMSO (Fisher Scientific) to 1 mg/mL as previously reported (Satou et al, 2004). The DiI solution was sprayed with a microneedle onto B7.6 cells of 64-cell stage embryos. Cytochalasin D (Sigma Aldrich, C8273), that was used to cleavage arrest of the

cells in ascidians (Hudson et al, 2003; Yamada, 2006; Karaiskou et al, 2015), was dissolved in DMSO and mixed with DiI in DMSO before use to make 2 mg/mL as a final concentration. These embryos were allowed to develop, fixed with MEM-FA or MEM-PFA (3.7% formaldehyde or 4% paraformaldehyde, 0.1 M MOPS, 0.5 M NaCl, 1 mM EGTA, 2 mM MgSO$_4$), and used for antibody staining and fluorescent in situ hybridization (FISH).

## Live imaging for lobe scission

Eggs and sperm were obtained from wildtype animals of *Ciona intestinalis*. The eggs were dechorionated and fertilized, and placed on an agarose-coated Petri dish. At 32-64 cell stage, B6.3 and B7.6 cells were sprayed with BioTracker MemBright 560 Live Cell Dye (Sigma-Aldrich, SCT084), that was diluted to 10 μM with PBS. The sprayed embryos were transferred to the agarose-coated Petri dish containing 100 nM BioTracker MemBright 488 Live Cell Dye (Sigma-Aldrich, SCT085), and kept 10-20 min. The stained embryos were put into the 35 mm culture dish with 15 mm glass (Navantor, VWR Confocal Dishes, 75856-744 ; VWR, 734-2905), that had mount of 1.5% agarose (Fisher Bioreagents, BP160-500) in ASW with graves made by a Glass Capillary 1 mm diameter (Narishige, GD-1). The z-stuck and t-stuck images were captured with the spinning disk confocal microscope (IXplore SpinSR, Olympus). The images were analyzed with Imaris software (Oxford instruments).

## Antibody staining

We used an antibody for RNA polymerase II as a previous report (Shirae-Kurabayashi et al, 2011); CTD-pSer2 (Abcam, ab5095, 1:500 dilution). We followed the previously described protocol (Ohta and Satou, 2013) with slight modification. A rabbit anti-human phospho-JAK2 (Y931) antibody (Thermo Fisher Scientific, PA5-104704) was used as a primary antibody, 1/500 in *Can Get Signal* Immunostain Solution A (TOYOBO). The antibody was detected by an anti-rabbit-HRP goat antibody 1/500 in Can Get Signal Immunostain Solution A (TOYOBO), and by Tyramide Signal Amplification (Perkin Elmer) as previously described (Ohta and Satou, 2013).

## In situ hybridization

DNA fragments were amplified by PCR with exTaq-HS (Takara Bio) and Phusion HF (New England Biolabs) DNA polymerases from *Ciona* genomic DNA or cDNA. The primers that we used were summarized in Appendix Table S1. The amplicons were subcloned into TOPO vectors (life technologies). DIG or fluorescein-labeled RNA probes were synthesized by T7 and sp6 RNA polymerases (Roche) from template DNA plasmid digested by NotI or SpeI (New England Biolabs), and were cleaned by RNeasy mini kit (QIAGEN). We followed the protocol for in situ hybridization described before (Christiaen et al, 2009b; Ohta and Satou, 2013). We detected fluorescein and DIG probes using TSA plus (Perkin Elmer), green (FP1168), and red (FP1170), respectively. Primer sequences are provided in Appendix Table S1.

We used an antibody for phospho histone 3 (pH3) that was previously reported (Shirae-Kurabayashi et al, 2006) (pH3-ser10-6g3-mouse-mAb, Cell Signaling; #9706; 1:500 diluted). The primary antibody was added together with anti-DIG antibody during FISH process, and secondary antibody (anti-mouse-Alexa-555, Thermo Scientific, A-21127)

was used to detect the anti-pH3 antibody after detection of ISH probe by using TSA plus (Perkin Elmer) green (FP1168).

## Pharmacological inhibitor treatments

Actinomycin D (50-76-0; A1410; Sigma) was diluted into DMSO at 10 mg/mL stock. The stock solution was diluted into ASW to a final concentration of 40 μg/mL. This concentration was reported to block transcription in *Halocynthia* embryos (Miyaoku et al, 2018). Flavopiridol (146426-40-6; S1230; Selleck chemicals) was diluted into water to 10 mM stock. The stock solution was diluted into ASW to final concentration 1 and 10 μM. The transcriptional inhibitor-treated embryos were fixed by MEM-PFA (4% PFA, 0.1 M MOPS, 0.5 M NaCl, 1 mM EGTA, 2 mM MgSO$_4$) after 1 h inhibitor treatment, and used for in situ hybridization.

1-Azakenpaullone (S7193; Selleckchem; Feinberg et al, 2019), Ruxolitinib (INCB018424; S1378; Selleckchem), Vismodegib (GDC-0449; S1082; Selleckchem), DAPT (208255-80-5; D5942; Millipore Sigma), SB431542 (S1067; Selleckchem; Ohta and Satou, 2013), U0126 (9903; Cell Signaling Technology; (Hudson et al, 2003)) and Dorsomorphin (1219168-18-9; S7306; Selleckchem; (Ohta and Satou, 2013; Feinberg et al, 2019)) were used to perturb define signaling pathways as described in corresponding references. These treatments were done in a final concentration of 10 μM for 2 or 4 h. The inhibitor-treated embryos were fixed by MEM-PFA after 2 h inhibitor treatment, and used for in situ hybridization.

## Vivo-morpholino oligonucleotide treatment

We designed antisense morpho oligonucleotide conjugated vivo (vivo-MO) at the 5' utr of *Jak-a* gene (5'-CTTTTGGTTAGCAT-GAATTGAAGCC-3'). The stock solution of vivo-MO was made by diluting with water to be 0.5 mM. Dechorionated eggs were treated with 20 μM, 40 μM and 60 μM for 20 min during 25–45 min after fertilization in 250 μL ASW in 2 mL microcentrifuge tube, and transferred into ASW on agarose coated Petri dishes until fixation with MEM-FA. Mock control embryos were treated with water the same volume of that of 60 μM vivo-MO condition. The fixed embryos were used for immunostaining. Because the previous report used vivo-MO in Zebrafish eggs and showed that 40 μM and 60 μM concentration of vivo-MO worked to block function of the target gene (Wong and Zohar, 2015), we tested the same concentration here.

## Statistical analysis

We used Microsoft Office Excel, R and R Studio to analyze our data, and analyzed the data by R and R Studio. The details of statistics were described in each figure legend. The standard error on the categorical data set was calculated by SE = SQRT(population*(1-population)/number). We used representative data on the figures after reproducing results in at least two independent experiments.

# Data availability

This study includes no data deposited in external repositories.

# Peer review information

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

## Acknowledgements

The authors thank Pr. Hiroki Nishida for helpful comments on the original version of this manuscript. The authors are grateful to Christiaen lab members for discussions and feedbacks. This work was supported by NIH/NIGMS award GM096032 and by core funding from the Michael Sars Centre to LC.

## Author contributions

**Naoyuki Ohta**: Conceptualization; Resources; Data curation; Software; Formal analysis; Validation; Investigation; Visualization; Methodology; Writing—original draft; Project administration; Writing—review and editing. **Lionel Christiaen**: Conceptualization; Supervision; Funding acquisition; Methodology; Writing—original draft; Project administration; Writing—review and editing.

## Funding

## Disclosure and competing interests statement

The authors declare no competing interests.

