## [Peer Review File · EMBO Reports]

Cellular remodeling and JAK inhibition promote zygotic gene expression in the *Ciona* germline

Naoyuki Ohta and Lionel Christiaen

Corresponding author(s): Naoyuki Ohta (naoyuki.ohta@uib.no), Lionel Christiaen (lionel.christiaen@uib.no)

Review Timeline:

Submission Date:	30th Jan 23
Editorial Decision:	21st Apr 23
Revision Received:	19th Jan 24
Editorial Decision:	15th Mar 24
Revision Received:	22nd Mar 24
Accepted:	28th Mar 24

Transaction Report:

Dear Naoyuki,

Thank you for the submission of your research manuscript to our journal. As you know, we have meanwhile received the full set of referee reports that is copied below.

As you will see, the referees acknowledge that the findings are potentially interesting, but they also raise a number of concerns, which need to be addressed during a revision. In particular:

(1) Lobe formation:

- All three referees ask for further data to substantiate the data on lobe formation. I think the suggestion to perform live imaging should be addressed experimentally, as it will address the concerns regarding better spatial and temporal resolution.
- There is a more general request for mechanism, maybe supported by scRNAseq data, which I feel goes beyond the current study and can be addressed in the point-by-point response and in the discussion.
- Inhibition of lobe formation to test whether it affects zygotic transcription: I think this concern should be addressed experimentally, maybe by using cytochalasin D, as suggested by referee 2.

(2) Tool validation (antibodies, JAK inhibitor): These concerns must be addressed experimentally, as suggested by the referees.

(3) Link between MEK/BMPR/GSK3 and JAK signalling and its more general effect on zygotic transcription beyond Mef2: These issues can be addressed textually, by appropriate and careful phrasing of the conclusions and by an in-depth discussion.

Taken together, we would like to invite you to revise your manuscript with the understanding that the referee concerns (as detailed above and in their reports) must be fully addressed and their suggestions taken on board. Please address all referee concerns in a complete point-by-point response. Acceptance of the manuscript will depend on a positive outcome of a second round of review. It is EMBO reports policy to allow a single round of revision only and acceptance or rejection of the manuscript will therefore depend on the completeness of your responses included in the next, final version of the manuscript.

We realize that it is difficult to revise to a specific deadline. In the interest of protecting the conceptual advance provided by the work, we recommend a revision within 3 months (July 21st). Please discuss the revision progress ahead of this time with the editor if you require more time to complete the revisions.

I am also happy to discuss the revision further via e-mail or a video call, if you wish.

*****IMPORTANT NOTE:

We perform an initial quality control of all revised manuscripts before re-review. Your manuscript will FAIL this control and the handling will be DELAYED if the following APPLIES:

- 1) A data availability section providing access to data deposited in public databases is missing. If you have not deposited any data, please add a sentence to the data availability section that explains that.
- 2) Your manuscript contains statistics and error bars based on $n=2$. Please use scatter blots in these cases. No statistics should be calculated if $n=2$.

When submitting your revised manuscript, please carefully review the instructions that follow below. Failure to include requested items will delay the evaluation of your revision.*****

- 1) a .docx formatted version of the manuscript text (including legends for main figures, EV figures and tables). Please make sure that the changes are highlighted to be clearly visible.
- 2) individual production quality figure files as .eps, .tif, .jpg (one file per figure). Please download our Figure Preparation Guidelines (figure preparation pdf) from our Author Guidelines pages <https://www.embopress.org/page/journal/14693178/authorguide> for more info on how to prepare your figures.

4) a complete author checklist, which you can download from our author guidelines (). Please insert information in the checklist that is also reflected in the manuscript. The completed author checklist will also be part of the RPF.

5) Please note that all corresponding authors are required to supply an ORCID ID for their name upon submission of a revised manuscript (). Please find instructions on how to link your ORCID ID to your account in our manuscript tracking system in our Author guidelines

()

6) We replaced Supplementary Information with Expanded View (EV) Figures and Tables that are collapsible/expandable online. A maximum of 5 EV Figures can be typeset. EV Figures should be cited as 'Figure EV1, Figure EV2' etc... in the text and their respective legends should be included in the main text after the legends of regular figures.

7) Please note that a Data Availability section at the end of Materials and Methods is now mandatory. In case you have no data that requires deposition in a public database, please state so instead of refereeing to the database. See also < <https://www.embopress.org/page/journal/14693178/authorguide#dataavailability>>. Please note that the Data Availability Section is restricted to new primary data that are part of this study.

Additional information on source data and instruction on how to label the files are available .

10) Figure legends and data quantification:

- the name of the statistical test used to generate error bars and P values,
- the number (n) of independent experiments (please specify technical or biological replicates) underlying each data point,
- the nature of the bars and error bars (s.d., s.e.m.)
- If the data are obtained from n {less than or equal to} 5, show the individual data points in addition to the SD or SEM.
- If the data are obtained from n {less than or equal to} 2, use scatter blots showing the individual data points.

11) Our journal encourages inclusion of *data citations in the reference list* to directly cite datasets that were re-used and obtained from public databases. Data citations in the article text are distinct from normal bibliographical citations and should directly link to the database records from which the data can be accessed. In the main text, data citations are formatted as follows: "Data ref: Smith et al, 2001" or "Data ref: NCBI Sequence Read Archive PRJNA342805, 2017". In the Reference list,

data citations must be labeled with "[DATASET]". A data reference must provide the database name, accession number/identifiers and a resolvable link to the landing page from which the data can be accessed at the end of the reference. Further instructions are available at .

12) As part of the EMBO publication's Transparent Editorial Process, EMBO reports publishes online a Review Process File to accompany accepted manuscripts. This File will be published in conjunction with your paper and will include the referee reports, your point-by-point response and all pertinent correspondence relating to the manuscript.

Kind regards,

Martina

Referee #1:

Lionel group's manuscript focused on the zygotic gene expression during early germline specification in *Ciona*. They first observed a unique cellular remodeling event that excluded Pem-1 mRNA. Then they found zygotic transcription begins based on the consecutive activation of RNAPII and Mef2 transcription. Finally, they discovered a potential antagonism between JAK and MEK/BMPR/GSK3 signaling that controls the timing of zygotic transcription initiation in the germline.

Overall, the manuscript proposed a new model for the onset of zygotic transcription in germline. The major results are reasonable, but several issues should be deeply addressed before the it could be accepted.

1. The cellular remodeling or the lobe structure process needs more detailed description, eg. what is the shape and size of the lobe? How was the lobe released (by exocytosis)? The driving mechanism of lobe formation is crucial, I think. Based on the mechanism, the authors could specifically stop lobe formation, then see what happened for zygotic gene expression.
2. The logic connection between cell remodeling and zygotic transcription need to be clearly built. The author only showed the temporal sequences between cell remodeling process (lobe scission) and zygotic transcription but it's not causal connection. Overexpression of pem-1 in PGCs might help to explain the effect of pem-1 exclusion in PGCs. In addition, large scale sequencing of single cell might be a power tool to resolve this problem and provide the convincing evidence on zygotic activation of germline.
3. Regarding the JAK inhibitor, does it specific? The knock-down of JAK will help to validate the JAK inhibitor experiment. Why did the authors only test the JAK signal in tailbud embryos? The JAK signal before lobe formation should also be tested. What is the possible mechanisms of JAK and MEK/BMPR/GSK3 signaling that controls the timing of zygotic transcription initiation? The authors should at least make some discussion.
4. The author claimed that "The endoderm helps PGC remodeling", but the results only showed the physical proximity. The "help" means the endoderm take part in this process actively. But I didn't find any cell contact evidence of endoderm cells.
5. Since the release of the inhibition of RNAPII from Pem-1 in PGCs, what about the expression of other genes except for mef-2?
6. Figure 4 was missing, it's the same with Figure 3 in manuscript.

Referee #2:

The current study consists of two parts: cellular remodeling of germline progenitor cells to exclude *PEM-1* transcripts and involvement of diverse signaling pathways in transcriptional activation in the germline lineage. I have a few comments on the first part of the study, which I find interesting and significant, while the second part requires more rigorous tests of the specificity of the tools used. I hope that the authors would find the comments below constructive and helpful:

Comments:

1) The cellular remodelling part would benefit from live imaging of B7.6 cells labelled with Dil and expressing fluorescent proteins targeted to different cellular compartments (cortex with Lifeact-GFP, nucleus with H2B-GFP etc). This can be done by injecting mRNAs encoding FPs into eggs and labelling B7.6 cells with Dil at the 64-cell stage. I am particularly interested in the position of lobe formation relative to the embryonic axis, how it is initiated, and how the lobe and the B7.6 become spatially separated along the tail.

2) It would be also neat if the authors could show that the exclusion of *PEM-1* transcripts via lobe formation is indeed important for the transcriptional activation in the germline lineage. Could it be possible to block the lobe formation by a brief treatment of embryos with cytochalasin D during pre- or post-lobe formation period and analyse its impact on pSer2 of the RNAPII-CTD and *Mef2* expression?

3) Ruxolitinib-treatment results in a loss of anti-P-JAK signals (Y971)(Fig. S4F). Is this expected?

4) More specificity tests need to be carried out on the tools used to address the role of JAK signals:

- How many JAK inhibitors were tested? What are resultant phenotypes like?

- What are the phenotypes of embryos treated with vivo-JAK-MO? Specifically, if the embryos gastrulate and reach to tailbud stages, do they exhibit the increase in *Mef2* transcription as in Ruxolitinib-treated embryos? And what about the anti-P-JAK signals in vivo-MO embryos at tailbud stage (st 21)?

- If the vivo-MO embryos fail to gastrulate, would a pre-gastrulation treatment of embryos with Ruxolitinib also result in a failure of gastrulation?

- Is it possible to treat gastrulating embryos with vivo-JAK-MO and test its impact on anti-P-JAK staining and *Mef2* expression at tailbud stages?

- I recommend the authors perform a western blot analysis to address the impact of vivo-JAK-MO treatment on P-JAK described in Fig. S5C.

- Finally, if the specificity turns out to be difficult to be proven, I would recommend removing the second part from the manuscript: I would support the amended manuscript, in particular, if it addresses the comments 1 and 2.

Minor comments:

1) The authors should describe how they identified B7.6s and their descendants when embryos were not Dil-labelled. It will be helpful for readers if the cells are labelled in each image (e.g., Fig. 2E; Fig. 3A'-D'; Fig. 3J,J').

2) In Fig S4A, I think that the tyrosine residue labelled with an asterisk is Y934 but not Y931.

3) All panels in Figure 4 are identical to those in Figure 3.

Referee #3:

This paper describes a previously unrecognized cell remodeling event of the PGC lineage in *Ciona robusta*, which plays a major role in removing a block on zygotic transcription via asymmetric distribution of *PEM-1*. The authors further show that interactions between several signaling pathways are responsible for initiating transcription in these B7.6* cells, although it is not clear where the MEK/BMPRI/GSK3 and JAK pathways are integrated- however, the authors clearly show it is not at the level of RNAPII phosphorylation or JAK phosphorylation. Thus there is an inverse correlation between *Mef2* expression and the proliferation of B7.6* cells shown in inhibitor studies, but it is not clear why this would be the case. Nevertheless, this is a solid study which is appropriate for this journal, and presents new antibodies and tools that are very valuable. I have some comments, but these are mostly minor issues in presentation which make the manuscript difficult to follow at times, and there are some important issues that should be introduced and discussed in a more comprehensive manner. Given the complexity of these experiments, I believe some re-writes would be helpful.

In Figure 1 and S1, it is very clear that the previously described division of the B7.6 cell is not a division, and I agree that this can be seen in the 2006 Shirae-Kurabayashi paper. The problem is that nowhere in the introduction nor the experiment is it stated when this should be happening, or why we are looking at stage 21. At this time point the lobe and B7.6* cell are several cell diameters apart. When did this scission happen? This information can be found later (Fig 4F and S1: stage 12) but it should be better described here so the figure can be understood. In addition, when is the B7.6* cell dividing? This is not introduced nor

discussed here either (again, it is later, but that is not helpful), but it should be, as Fig 1G and G' clearly show two adjacent cells that are Dil labeled, at stage 16. I found myself going back and forth in the document and Figures to understand what was going on, and thus believe it should be better introduced and described. Also, why is the lobe still present several hours later? This is not discussed either.

Minor: on the bottom of page 3, sentence reads '...eggs and early embryos, lobe tended to segregate...' should read 'the lobe'.

The section labeled: The endoderm helps PGC remodeling is unclear. I'm not sure what 'help' means- it is clear that this process occurs near the endoderm but there is no data that the endoderm is helping anything. This should be re-written. Also, are figures 1M and 1N the same animal, I cannot figure out the orientation, and the large cell on the bottom of panel M is not seen in panel N.

The use of Mef2 was also a bit confusing. The data with the intronic probes is great, but I don't understand how it was identified as a candidate if it so much of it is maternally deposited. I may be missing something here. Also, when does the Actinomycin D and Flavopiridol treatments start? It is not clear from text or figure legends.

Using Mef2 intronic expression as a proxy for zygotic transcription, the authors next ask what other inputs would control zygotic expression following the loss of PEM-1 dependent RNAPII inhibition. Blocking the MEK1/2, BMPRI and GSK3 pathways independently do not have a major affect, but the combination of all three does. While it is not clear exactly what is going on (which pathways contribute, how it would be additive, etc.), it is not surprising that this would be a very complex process. The JAK inhibitor increases Mef2 expression, thus these four pathways are clearly affecting transcription of this transcript in a positive (3i) or negative (JAK) manner, respectively. The only issue here is that it is not clear if this is a general effect on transcription or specific to Mef2. Mef2 was picked as a proxy for transcription, as such it is unlikely that 4 inhibitors are having a specific effect on that gene, but it is certainly possible and should be discussed. In addition, like any of the '3i' pathways, JAK is a signal transduction molecule downstream of some receptor. Is that known? Is there any hint from the scRNA seq data set published several years ago?

In addition, where do the authors think that the signals from the '3i' pathways and the JAK pathway are integrated, given that this is clearly not happening at POLII phosphorylation or JAK phosphorylation? This should be commented on.

Finally, there is a negative correlation between Mef-2 transcription and proliferation. Is this change in proliferation due to a general inhibition of transcription? This is not specifically discussed either, which seems strange given that it is a very important point for this study.

In summary, I think this is a good study but believe there are some deficiencies in both the introduction, description of experiments and discussion. The data supporting the remodeling vs. cell division is very clear, while the licensing and positive and negative regulation of zygotic transcription following the loss of PEM-1 is not as clear, but certainly has opened the door for further studies. As such, while I think the study could be more clearly and thoroughly described and discussed, it will be suitable for publication in EMBO following revisions.

Dear Naoyuki,

Thank you for the submission of your research manuscript to our journal. As you know, we have meanwhile received the full set of referee reports that is copied below.

As you will see, the referees acknowledge that the findings are potentially interesting, but they also raise a number of concerns, which need to be addressed during a revision. In particular:

(1) Lobe formation:

- All three referees ask for further data to substantiate the data on lobe formation. I think the suggestion to perform live imaging should be addressed experimentally, as it will address the concerns regarding better spatial and temporal resolution.

In the previous versions, we described the dynamics of lobe formation from fixed samples every 30 minutes (Figure S1). In this revised version, we performed live imaging to complement the analysis. Live imaging of internal structures is notoriously difficult with early *Ciona* embryos, which are NOT transparent, unlike the embryos of other species such as *Phallusia mammillata*. This is the main reason why it took a long time to acquire interpretable data from live specimens. We eventually managed to record time lapse data showing lobe formation from B7.6 cells labeled in situ with the chemical dye BioTracker MemBright Live Cell Dye. We provide supplemental movies S1 and S2, and more easily interpretable snapshots in main Figure 1.

- There is a more general request for mechanism, maybe supported by scRNAseq data, which I feel goes beyond the current study and can be addressed in the point-by-point response and in the discussion.

We wish to argue that the data and conclusions presented contain a substantial amount of mechanistic data, even though there is – as often – room for more. We wish to stress that, given the standards of the field of developmental studies using *Ciona* embryo, we provide a respectable amount of new insights and technical innovations. We do have our own whole embryo scRNA-seq data, which indeed goes well beyond the scope of this manuscript. We will thus focus on point-by-point responses as suggested.

- Inhibition of lobe formation to test whether it affects zygotic transcription: I think this concern should be addressed experimentally, maybe by using cytochalasin D, as suggested by referee 2. This is a great point, which we attempted unsuccessfully by inhibiting endoderm function, but not with cytochalasin treatment. In this revised version, we used a microneedle to spray Cytochalasin D specifically onto B7.6 cells, which blocked lobe formation and caused *Pem-1* mRNA to disperse throughout the cytoplasm of PGCs. We added these results in Figure 1. This challenging experiment, which, to our knowledge, has never been performed using *Ciona* embryos, limits the number of samples that could be obtained for subsequent evaluation of the transcriptional activity using the anti RNAP CTD-pSer2 antibody. Because we observed remaining signal in cytochalasin inhibited cells, we cannot formally conclude based on these experiments and reasoned that additional mechanisms, presumably involving targeted *Pem-1* protein degradation, might complement lobe formation and mRNA exclusion to inhibit *Pem-1* in

a timely manner. In the future, we plan to manage to combine perturbation of protein degradation in B7.6* cells, with perturbation of lobe formation and assay the dynamics of Pem-1 protein abundance, RNAP activity and zygotic genome activation. The additional experiments needed to fully address these points go well beyond the scope of this paper. To prevent further delay, the first version of this paper having been posted on BioRxiv in 2021, we chose to discuss and mention the potential role of Pem-1 protein degradation in both text and figures, to prevent implications that segregation of Pem-1 RNA is only one critical step to remove Pem-1 gene products and license zygotic gene expression in PGCs.

(2) Tool validation (antibodies, JAK inhibitor): These concerns must be addressed experimentally, as suggested by the referees.

We provided supplemental data aimed specifically at testing the specificity of both the anti pJAK and Ruxolitinib inhibitor, and back these evidence with antisense morpholino oligonucleotide-mediated knock-down of the endogenous maternal Jak-a, which inhibited the anti pJAK staining. In the absence of acknowledgement of the specific controls that we present, and the relatively general and vague criticisms, it is rather difficult to know exactly what the reviewers would expect to be done. Importantly, the alternative explanations that both the morpholino, antibody and inhibitor cause non-specific effects seems unlikely.

(3) Link between MEK/BMPR/GSK3 and JAK signalling and its more general effect on zygotic transcription beyond Mef2: These issues can be addressed textually, by appropriate and careful phrasing of the conclusions and by an in-depth discussion.

This is a good point, which we obviously considered and would address experimentally using perturbations combined with single cell genomics, which would undeniably expand the scope of the paper beyond the reasonable frame of the current manuscript.

Taken together, we would like to invite you to revise your manuscript with the understanding that the referee concerns (as detailed above and in their reports) must be fully addressed and their suggestions taken on board. Please address all referee concerns in a complete point-by-point response. Acceptance of the manuscript will depend on a positive outcome of a second round of review. It is EMBO reports policy to allow a single round of revision only and acceptance or rejection of the manuscript will therefore depend on the completeness of your responses included in the next, final version of the manuscript.

We realize that it is difficult to revise to a specific deadline. In the interest of protecting the conceptual advance provided by the work, we recommend a revision within 3 months (July 21st). Please discuss the revision progress ahead of this time with the editor if you require more time to complete the revisions.

I am also happy to discuss the revision further via e-mail or a video call, if you wish.

You can either publish the study as a short report or as a full article. For short reports, the

revised manuscript should not exceed 27,000 characters (including spaces but excluding materials & methods and references) and 5 main plus 5 expanded view figures. The results and discussion sections must further be combined, which will help to shorten the manuscript text by eliminating some redundancy that is inevitable when discussing the same experiments twice. For a normal article there are no length limitations, but it should have more than 5 main figures and the results and discussion sections must be separate. In both cases, the entire materials and methods must be included in the main manuscript file.

*****IMPORTANT NOTE:

We perform an initial quality control of all revised manuscripts before re-review. Your manuscript will FAIL this control and the handling will be DELAYED if the following APPLIES:

1) A data availability section providing access to data deposited in public databases is missing. If you have not deposited any data, please add a sentence to the data availability section that explains that.

2) Your manuscript contains statistics and error bars based on $n=2$. Please use scatter blots in these cases. No statistics should be calculated if $n=2$.

We wish to indicate that error bars on barplots representing categorical data do not show the spread across replicates, but the standard error of a proportion (SEP), which depends on the proportion p , and the number of individual observations n as follows: $SEP = \sqrt{p * (1-p) / n}$. this is a statistically valid way to represent the spread of a proportion estimate, and does not represent a p-value. We indicated this in the figure legends and methods section.

Referee #1:

Lionel group's manuscript focused on the zygotic gene expression during early germline specification in Ciona.

They first observed a unique cellular remodeling event that excluded Pem-1 mRNA. Then they found zygotic transcription begins based on the consecutive activation of RNAPII and Mef2 transcription. Finally, they discovered a potential antagonism between JAK and MEK/BMPR/GSK3 signaling that controls the timing of zygotic transcription initiation in the germline.

Overall, the manuscript proposed a new model for the onset of zygotic transcription in

germline. The major results are reasonable, but several issues should be deeply addressed before the it could be accepted.

1. The cellular remodeling or the lobe structure process needs more detailed description, eg. what is the shape and size of the lobe?

A: We agree with the reviewer that documenting these details would be nice, especially if lobe formation could be readily observed in 30+ live embryos per experiment, for accurate measurements from live imaging data. This is currently not feasible, and since we do not attempt to draw formal conclusions from the shape and size of the lobe, it is our opinion that such questions shall be addressed in future studies.

How was the lobe released (by exocytosis)?

A: Likewise, we share the reviewer's interest in the specific mechanism, but did consider this to be a follow-up question, as we focused on a rather bird's eye view of a new phenomenon. This said, exocytosis is not compatible with the observed topology. Rather, the lobe seems to first forms as a protrusion, which would make its scission more comparable to apocrine secretion. Unlike the analogous process observed in *C. elegans*, we do not think this is *trogocytosis*, as the endoderm cells do not seem to "bite off" the germline lobe. In this revised version, we added B7.6-specific cytochalasin treatment and preliminary live imaging supporting the notion that the lobe forms through actin-mediated protrusive activity.

The driving mechanism of lobe formation is crucial, I think.

A: The mechanism is of high interest indeed, but is it crucial to solve it in order to demonstrate the existence of the phenomenon? Our paper focuses squarely on demonstrating the phenomenon of *Pem-1* RNA exclusion by lobe formation and scission. We stand by our conclusion that this happens in ascidian embryos, and agree that we do not have a full mechanism, except for the new cytochalasin D data, but we argue that does not diminish the value of our data in supporting the conclusion that the phenomenon exists.

There is no doubt that adding mechanistic insights would augment the scientific value of the paper. But it would not change its validity. Augmenting insights would also extend the scope of this one and first paper on the topic. It is rare for scientific papers to both report on the discovery of a new phenomenon, and solve the mechanistic underpinnings of how this happens. We confess being somewhat dismayed by what we interpret as taking the novelty of the phenomenon for granted, while focusing on questions that extend beyond the scope of the paper. On the other hand, we are glad to see that our findings trigger such excitement and curiosity, which we share with the reviewers.

Based on the mechanism, the authors could specifically stop lobe formation, then see what happened for zygotic gene expression.

A: We agree with the reviewer that this is important, and squarely part of the study and the final model. In this revised version, we sprayed cytochalasin D together with Dil specifically onto B7.6 cells to inhibit actin polymerization, and both lobe formation and monitored *Pem-1* RNA

distribution. Note that such cell-specific cytochalasin treatment is unprecedented in the field, and quite tedious. Our new data indicate that the lobe scission requires actin dynamics, which was expected, as does *Pem-1* RNA concentration and accumulation in the prospective lobe. Further combining these assays with anti RNAP CTD pSer2 immunostaining is very challenging, and we could not connect the blocking of lobe formation to initiation of the zygotic gene expression more directly in this version. We amended our final model and discussion to acknowledge the possibility that targeted protein degradation could provide a complementary mechanism to remove *Pem-1* gene products from the PGCs and license zygotic gene expression. Fully addressing the potential role of post-translation regulation of *Pem-1* extends beyond the scope of this study.

2. The logic connection between cell remodeling and zygotic transcription need to be clearly built.

We agree with the reviewer and this is essentially the same point as above: block lobe formation and assay the onset on zygotic transcription, specifically RNAPII activity with anti RNAP-CTD phospho-Serine2 immunostaining. In this revised version, we managed to inhibit lobe formation and scission using a novel cell-specific cytochalasin treatment. We showed that this inhibits lobe formation and caused *Pem-1* RNA to redistribute throughout the PGC cytoplasm. In order to dig this question deeper, we plan to establish ways to block protein degradation in B7.6* cells and detect *Pem-1* protein presence and degradation. Preliminary RNAP CTD pSer2 immunostaining suggested the existence of additional, post-transcriptional and post-translational, layers of *Pem-1* regulation, which will need to be addressed in future work. We acknowledge this is our revised discussion and summary model.

The author only showed the temporal sequences between cell remodeling process (lobe scission) and zygotic transcription but it's not causal connection. Overexpression of *pem-1* in PGCs might help to explain the effect of *pem-1* exclusion in PGCs.

To this point, we cited previously published work showing that overexpression of *Pem-1* blocks zygotic transcription. This is consistent with our conclusion that removing *Pem-1* is necessary to license RNAPII activity. Here, we focused on blocking lobe formation, and assay *Pem-1* RNA distribution. Preliminary RNAP CTD pSer2 immunostaining suggested the existence of additional, post-transcriptional and post-translational, layers of *Pem-1* regulation, which will need to be addressed in future work. We acknowledge this is our revised discussion and summary model.

In addition, large scale sequencing of single cell might be a power tool to resolve this problem and provide the convincing evidence on zygotic activation of germline.

A: We agree with the reviewer that scRNA-seq over a time course is a fantastic approach to uncover cell-type specific gene expression. Of note, a published whole embryo scRNA-seq study briefly suggested that the germline remains transcriptionally silent during embryogenesis (Cao et al., Nature, 2019). Our unpublished whole embryo scRNA-seq time course provides a different view, more in line with an onset of zygotic gene expression as described in this paper.

Hopefully, the reviewer will accept the notion that a thorough reporting of this dataset extends well beyond this manuscript, and – more importantly – that we provide experimental evidence for the conclusions that we put forward in the paper at hand.

3. Regarding the JAK inhibitor, does it specific?

A: We agree with the reviewer that asserting the efficacy and specification of drug treatments is essential and unfortunately often incompletely reported in the field. Specifically, in this and previous version we provided several lines of evidence supporting the notion that the Ruxolitinib treatment impacts JAK signaling in *Ciona* embryos, as follows:

- this inhibitor is known to be specific to JAK signaling in other systems;
- we identified a conserved peptide containing a phospho-tyrosine between *Ciona* Jak.a and human JAK2
- We performed immunostaining using a polyclonal antibody raised against the same peptide in human JAK2 that is conserved in *Ciona*.
- We showed that the inhibitor blocks staining by this anti phospho-JAK2
- We showed that the same staining with anti phospho-JAK2, which we report in early embryos, is also blocked by a *vivo* morpholino targeting the endogenous Jak.a
- And we showed that Jak.a mRNA are maternally loaded in early embryos.

We would like to argue that the above bundle of evidence strongly supports the notion that both the inhibitor treatment and anti-pJAK immunostaining are specific to endogenous JAK activity, as discussed on page 6 line 29 to page 7 line 21 and page 8 line 16-20. Assertion to the contrary should be argued, in part by providing plausible explanations to our observations in the context of non-specific effects.

The knock-down of JAK will help to validate the JAK inhibitor experiment.

A: We performed this experiment and assayed pJAK in this and previous versions of the paper as mentioned above.

Why did the authors only test the JAK signal in tailbud embryos? The JAK signal before lobe formation should also be tested.

A: We performed this experiment and assayed pJAK in this and previous versions of the paper, as shown in the supplemental figure S6.

What is the possible mechanisms of JAK and MEK/BMPR/GSK3 signaling that controls the timing of zygotic transcription initiation? The authors should at least make some discussion.

A: We agree with the reviewer that this interesting question. We wish to argue that addressing these points would also extend well beyond the scope of this first study. We attempted to focus discussion points on the results, and avoid speculation that would be too removed from our observations.

4. The author claimed that "The endoderm helps PGC remodeling", but the results only showed

the physical proximity. The "help" means the endoderm take part in this process actively. But I didn't find any cell contact evidence of endoderm cells.

A: We agree with the reviewer that we no longer show any effect of the endoderm on lobe formation and/or scission, contrary to the initial preprint version that contained preliminary evidence to it. We removed this assertion and focus on the spatial proximity/physical association in order to discuss the contrast with *C. elegans*, where intestinal progenitors engulf the lobe (trogocytosis).

5. Since the release of the inhibition of RNAPII from Pem-1 in PGCs, what about the expression of other genes except for *mef-2*?

A: Similar to our answer to point 2, we agree that this is a very interesting question, but also wish to point out that we refrained from speculating about it in the absence of genome-wide data. We wish to humbly request to focus the evaluation to the conclusions drawn from the results presented, as we could not possibly address all the remaining questions that emerge from our initial findings.

6. Figure 4 was missing, it's the same with Figure 3 in manuscript.

A: We apologize for this oversight and replaced the correct Figure.

Referee #2:

The current study consists of two parts: cellular remodeling of germline progenitor cells to exclude *PEM-1* transcripts and involvement of diverse signaling pathways in transcriptional activation in the germline lineage. I have a few comments on the first part of the study, which I find interesting and significant,

while the second part requires more rigorous tests of the specificity of the tools used.

I hope that the authors would find the comments below constructive and helpful:

Comments:

1) The cellular remodelling part would benefit from live imaging of B7.6 cells labelled with Dil and expressing fluorescent proteins targeted to different cellular compartments (cortex with Lifeact-GFP, nucleus with H2B-GFP etc). This can be done by injecting mRNAs encoding FPs into eggs and labelling B7.6 cells with Dil at the 64-cell stage.

A: We agree with the reviewer that extensive live imaging of lobe formation would be fantastic. While we are not aware of published work performing experiments analogous to the one suggested, we wish to share our experience that (1) Dil gets internalized very quickly and is difficult to use for live imaging and (2) the rather opaque early *Ciona* embryos (by contrast with transparent *Phallusia mammillata*) make these experiments very challenging, which partially explain the long delay of our resubmission. Here, we employed an alternative method, using the BioTracker MemBright family of chemical dyes to label cell membrane in two colors to

distinguish the whole embryo (green) from B7.6 cells (red), by spraying the red dye onto B7.6 cells around the 64-cell stage, and image live embryos with a spinning disk confocal microscope. These experiments remain extremely challenging, but we provide 2 movies as supplemental movies, and clearer snapshots showing lobe formation in main Figure 1 I-J.

I am particularly interested in the position of lobe formation relative to the embryonic axis, how it is initiated, and how the lobe and the B7.6 become spatially separated along the tail.

A: We agree with the reviewer that the cell biology of lobe formation is fascinating and are grateful for their curiosity and enthusiasm. We are hopeful that further insights will be gained through future studies by us and/or others.

2) It would be also neat if the authors could show that the exclusion of *PEM-1* transcripts via lobe formation is indeed important for the transcriptional activation in the germline lineage.

Could it be possible to block the lobe formation by a brief treatment of embryos with cytochalasin D during pre- or post-lobe formation period and analyze its impact on pSer2 of the RNAPII-CTD and *Mef2* expression?

A: We fully agree with the reviewer that this is an important point. In this revised version we blocked lobe formation using a novel blastomere-specific cytochalasin D treatment, using a microneedle to spray a mixture of Dil and cytochalasin specifically onto B7.6 cells. This tedious treatment blocked lobe formation and caused *Pem-1* RNA scatter through the cytoplasm. This data indicates that the lobe scission requires actin dynamics, as expected, as does *Pem-1* RNA concentration, but we stopped short of further connecting the blocking of lobe formation to initiation of the zygotic gene expression.

We acknowledged this limitation by amending the discussion and summary figure (Figure 4G) to open the possibility, which we think is real, that additional post-transcriptional and post-translational layers of *Pem-1* regulation complement RNA exclusion by lobe scission to remove *Pem-1* gene products from the PGCs, and license zygotic gene expression.

We wish to argue that a thorough experimental analysis of *Pem-1* post-transcriptional and post-translational regulation in PGCs extends well beyond the scope of this paper.

3) Ruxolitinib-treatment results in a loss of anti-P-JAK signals (Y971)(Fig. S4F). Is this expected?

A: Yes indeed. JAK is a kinase that auto-/cross-phosphorylates (e.g. Lodish et al., Molecular Cell Biology, 8th Ed., Ch. 16.2, pp. 726-730, and Fig. 16-10), such that inhibiting JAK activity should also inhibit JAK phosphorylation.

4) More specificity tests need to be carried out on the tools used to address the role of JAK signals:

A: We agree with the reviewer that asserting the efficacy and specification of drug treatments is essential and unfortunately often incompletely reported in the field. Specifically, in this and previous version we provided several lines of evidence supporting the notion that the Ruxolitinib treatment impacts JAK signaling in *Ciona* embryos, as follows:

- this inhibitor is known to be specific to JAK signaling in other systems;
- we identified a conserved peptide containing a phospho-tyrosine between *Ciona* Jak.a and human JAK2
- We performed immunostaining using a polyclonal antibody raised against the same peptide in human JAK2 that is conserved in *Ciona*.
- We showed that the inhibitor blocks staining by this anti phospho-JAK2
- We showed that the same staining with anti phospho-JAK2, which we report in early embryos, is also blocked by a vivo morpholino targeting the endogenous Jak.a
- And we showed that Jak.a mRNA are maternally loaded in early embryos.

We would like to argue that the above bundle of evidence strongly supports the notion that both the inhibitor treatment and anti-pJAK immunostaining are specific to endogenous JAK activity, as discussed on page 6 line 29 to page 7 line 21 and page 8 line 16-20. Assertion to the contrary should be argued, in part by providing plausible explanations to our observations in the context of non-specific effects.

- How many JAK inhibitors were tested? What are resultant phenotypes like?

A: We appreciate the general suggestion and wonder which other inhibitors should we have tested? We assume that the reasoning is that testing other inhibitors would presumably control for the specificity as non-specific side effects would likely be different. We would like to argue that we provided controls for the specificity as detailed above, and that the alternative hypothesis, that the drug has non-specific effects despite all the consistent observations listed, is very unlikely.

- What are the phenotypes of embryos treated with vivo-JAK-MO? Specifically, if the embryos gastrulate and reach to tailbud stages, do they exhibit the increase in *Mef2* transcription as in Ruxolitinib-treated embryos? And what about the anti-P-JAK signals in vivo-MO embryos at tailbud stage (st 21)?

A: These are great questions and indeed we found that the impact of vivo-MOs was diluted over time, which prevented us from addressing these questions, which is why we focused on drug treatments that provide an additional layer of (temporal) control. Of note, vivo-MO are not commonly used in *Ciona*, our results thus provide a novel precedent to build upon in future studies.

- If the vivo-MO embryos fail to gastrulate, would a pre-gastrulation treatment of embryos with Ruxolitinib also result in a failure of gastrulation?

A: We did not show such data. Indeed, we observed the late stage of the embryo treated with vivo-MO of jak, the larvae looked fine seemed to have processed normal gastrulation.

- Is it possible to treat gastrulating embryos with vivo-JAK-MO and test its impact on anti-P-JAK staining and *Mef2* expression at tailbud stages?

A: We appreciate the suggestion and would like to emphasize that Jak-a is maternally deposited, which makes it likely that a later treatment would still allow for early protein production and

thus further diminish the knock-down effect. While all of this remains in principle testable, we deemed the cost-benefit ratio of such experiment to be too high to warrant further delay.

- I recommend the authors perform a western blot analysis to address the impact of vivo-JAK-MO treatment on P-JAK described in Fig. S5C.

A: We performed an immunohistochemical staining to address this point, and would like to argue that the quantity of material required to perform the suggested experiment renders it rather prohibitive, not to mention the fact that we would lose the spatial resolution present in the in situ assay.

- Finally, if the specificity turns out to be difficult to be proven, I would recommend removing the second part from the manuscript: I would support the amended manuscript, in particular, if it addresses the comments 1 and 2.

A: We wish to argue that the second part of the manuscript is an essential part of it, which presents several novel observations, and with controls that exceed those usually reported in studies using *Ciona* for developmental biology. As stated above, the possibility that the drug treatment and anti phospho-JAK staining are non-specific is largely inconsistent with the bundle of evidence presented.

Minor comments:

1) The authors should describe how they identified B7.6s and their descendants when embryos were not Dil-labelled. It will be helpful for readers if the cells are labelled in each image (e.g., Fig. 2E; Fig. 3A'-D'; Fig. 3J,J').

A: We used the B7.6 and PGC marker *Ms4a/Pem7* for this is indicated repeated in the text and figures.

2) In Fig S4A, I think that the tyrosine residue labelled with an asterisk is Y934 but not Y931.

A: We are grateful for this correction, which we implemented in the revised version.

3) All panels in Figure 4 are identical to those in Figure 3.

A: We replaced with the correct Figure.

Referee #3:

This paper describes a previously unrecognized cell remodeling event of the PGC lineage in *Ciona robusta*, which plays a major role in removing a block on zygotic transcription via asymmetric distribution of PEM-1. The authors further show that interactions between several signaling pathways are responsible for initiating transcription in these B7.6* cells, although it is

not clear where the MEK/BMPRI/GSK3 and JAK pathways are integrated- however, the authors clearly show it is not at the level of RNAPII phosphorylation or JAK phosphorylation.

Thus there is an inverse correlation between Mef2 expression and the proliferation of B7.6* cells shown in inhibitor studies, but it is not clear why this would be the case.

We would like to state that this concept of inverse correlation between Mef2 expression and proliferation of B7.6 is a terminology used here by the reviewer, but we are not sure what this refers to.

Nevertheless, this is a solid study which is appropriate for this journal, and presents new antibodies and tools that are very valuable. I have some comments, but these are mostly minor issues in presentation which make the manuscript difficult to follow at times, and there are some important issues that should be introduced and discussed in a more comprehensive manner. Given the complexity of these experiments, I believe some re-writes would be helpful.

In Figure 1 and S1, it is very clear that the previously described division of the B7.6 cell is not a division, and I agree that this can be seen in the 2006 Shirae-Kurabayashi paper. The problem is that nowhere in the introduction nor the experiment is it stated when this should be happening, or why we are looking at stage 21. At this time point the lobe and B7.6* cell are several cell diameters apart. When did this scission happen? This information can be found later (Fig 4F and S1: stage 12) but it should be better described here so the figure can be understood.

A: We agree with the reviewer that this point could be clarified. We added new panels and data, including provisional live imaging and time series, to more clearly show that lobe formation and scission happen during gastrulation.

In addition, when is the B7.6* cell dividing? This is not introduced nor discussed here either (again, it is later, but that is not helpful), but it should be, as Fig 1G and G' clearly show two adjacent cells that are Dil labeled, at stage 16. I found myself going back and forth in the document and Figures to understand what was going on, and thus believe it should be better introduced and described. Also, why is the lobe still present several hours later? This is not discussed either.

A: We are grateful for these suggestions to improve the readability of the manuscript. We now describe when B7.6* cells divide in Figure 1D.

Minor: on the bottom of page 3, sentence reads '....eggs and early embryos, lobe tended to segregate...' should read 'the lobe'.

A: Yes indeed we agree and went again through the manuscript to address these points.

The section labeled: The endoderm helps PGC remodeling is unclear. I'm not sure what 'help' means- it is clear that this process occurs near the endoderm but there is no data that the endoderm is helping anything. This should be re-written. Also, are figures 1M and 1N the same animal, I cannot figure out the orientation, and the large cell on the bottom of panel M is not seen in panel N.

A: We agree and acknowledge that these statements were leftovers from previous versions that contain data on the endoderm too preliminary to be included in the new versions of the manuscript. We now simply describe the proximity and association of the lobe and germ cells with the endoderm in order to compare with *C. elegans* and discuss differences. The figures 1M and 1N in previous version moved to Supplemental figure 2, and we added captions into these figures to clarify the orientations of embryos and place of lobe&B7.6* cells.

The use of Mef2 was also a bit confusing. The data with the intronic probes is great, but I don't understand how it was identified as a candidate if so much of it is maternally deposited. I may be missing something here.

A: The reviewer is correct that we found it by serendipity in ways that would be too convoluted to recount the history of this finding. The bottomline is that we now show data on this gene, and only conclude from the data we show, but not attempt to recount the random observations that led us to test it.

Also, when does the Actinomycin D and Flavopiridol treatments start? It is not clear from text or figure legends.

A: One hour before fixation, as written in the Methods section.

Using Mef2 intronic expression as a proxy for zygotic transcription, the authors next ask what other inputs would control zygotic expression following the loss of PEM-1 dependent RNAPII inhibition. Blocking the MEK1/2, BMPRI and GSK3 pathways independently do not have a major affect, but the combination of all three does. While it is not clear exactly what is going on (which pathways contribute, how it would be additive, etc.), it is not surprising that this would be a very complex process. The JAK inhibitor increases Mef2 expression, thus these four pathways are clearly affecting transcription of this transcript in a positive (3i) or negative (JAK) manner, respectively. The only issue here is that it is not clear if this is a general effect on transcription or specific to Mef2.

A: We agree with the review and this is a good point. We attempt to carefully word our conclusions to strictly adhere to the data presented, and indeed cannot rule out that a number of zygotically expressed germline genes are insensitive to JAK signaling and/or either of the other pathways.

Mef2 was picked as a proxy for transcription, as such it is unlikely that 4 inhibitors are having a specific effect on that gene, but it is certainly possible and should be discussed. In addition, like any of the '3i' pathways, JAK is a signal transduction molecule downstream of some receptor. Is that known?

A: We discussed the parallel with mammalian stem cells at the very end of the discussion, which also rely on inhibiting MAPK, GSK3 and BMP and maintain JAK to preserve stemness.

Is there any hint from the scRNA seq data set published several years ago?

A: With regards to scRNA-seq, the Cao et al. paper briefly suggests that there is no zygotic transcription in the germline. Our own unpublished whole embryo scRNA-seq time course (in

preparation) indicates otherwise, but including it would be well beyond the scope of this manuscript, which - as we stress throughout this rebuttal - has a defined focus.

In addition, where do the authors think that the signals from the '3i' pathways and the JAK pathway are integrated, given that this is clearly not happening at POLII phosphorylation or JAK phosphorylation? This should be commented on.

A: As mentioned above, we briefly discussed a parallel with mammalian stem cells, but we also sought to keep the speculation to a minimum and to focus the discussion on the main conclusions that were supported by our observations.

Finally, there is a negative correlation between Mef-2 transcription and proliferation. Is this change in proliferation due to a general inhibition of transcription? This is not specifically discussed either, which seems strange given that it is a very important point for this study.

A: We do not specifically call it a negative correlation as we have too few conditions that affect both readouts independently. The effects of JAK inhibition on Mef2 expression and on cell division could be totally independent, as suggested by the precocious upregulation of Mef2 in cells that have not divided. We are puzzled by these comments as we did not attempt to draw any conclusion from a potential relationship with Mef2 expression and PGC divisions.

In summary, I think this is a good study but believe there are some deficiencies in both the introduction, description of experiments and discussion. The data supporting the remodeling vs. cell division is very clear, while the licensing and positive and negative regulation of zygotic transcription following the loss of PEM-1 is not as clear, but certainly has opened the door for further studies. As such, while I think the study could be more clearly and thoroughly described and discussed, it will be suitable for publication in EMBO following revisions.

A: Thank you so much.

Dear Lionel,

Thank you for the submission of your revised manuscript to EMBO reports. It has been seen again by referee #2 who considers your response to the referee concerns adequate and supports publication of your study in EMBO Reports (see report below).

Browsing through the manuscript myself, I noticed a few editorial things that we need before we can proceed with the official acceptance of your study.

- Your manuscript will be published as Scientific Report. We therefore ask you to please combine the Results and Discussion sections.

- Please submit the manuscript text as a .docx formatted version that includes legends for main figures, EV figures and tables. Please do not use a column format.

- We need the figures as individual production quality figure files (as .eps, .tif, .jpg; one file per figure). Please download our Figure Preparation Guidelines (figure preparation pdf) from our Author Guidelines pages <https://www.embopress.org/page/journal/14693178/authorguide> for more info on how to prepare your figures.

- Please add up to 5 keywords on the title page.

- Please add a 'Disclosure and competing interests statement'. For more information see <https://www.embopress.org/page/journal/14693178/authorguide#conflictsofinterest>

- Please note the following order of the manuscript sections: Title page - Abstract - Introduction - Results - Discussion - Methods - Acknowledgements - Disclosure and competing interests statement - References - Figure legends - Tables and their legends (not EV tables) - Expanded View Figure legends

- References: DOIs should only be used for preprints and datasets that have not been published yet. The [PREPRINT] label for Imai et al, 2004 should be removed from the reference list.

- Please add callouts for Figure 4F and Table S1 and S2.

- Supplementary information: You have currently 8 Supplementary figures. You could combine these with their legends into one PDF file called "Appendix". The Appendix needs a title page with a table of contents (incl. page numbers) and the nomenclature is "Appendix Figure S#". Alternatively, you could promote up to 5 Figures to the Expanded View content (displayed within the HTML in a collapsible format), the rest would need to go to the above mentioned Appendix. If you choose the EV option, the nomenclature is "Figure EV#" and the legends are part of the manuscript file as "Expanded View Figure Legends" following the main figure legends.

- Movies: please provide legends for the movies in the form of a readme.txt file; each movie should be zipped up together with its legend; the correct nomenclature and callouts are Movie EV1, Movie EV2.

- Please provide a complete author checklist, which you can download from our author guidelines (<<https://www.embopress.org/page/journal/14693178/authorguide>>). Please insert information in the checklist that is also reflected in the manuscript. The completed author checklist will also be part of the RPF.

- Funding information: "core funding from the Michael Sars Centre to L.C." needs to be removed from the Comments box in our manuscript tracking system and inserted as a funder via "More Funders options".

- Source data: the files and folders are perfectly fine in order but please upload each figure folder separately. Currently, you have them all zipped together. Thank you.

The .xls file should also be provided as one .xls file per figure (including all the panels in the same .xls file) and included in the folder.

- Table S1 and Table S2 need legends - if they are supplementary tables, they should also be in the Appendix file as Appendix Table S1 and S2; if they are main tables they should be in the manuscript file between main and EV figure legends and labeled as Table 1 and 2; if Expanded View, then they should remain as separate files but the names and callouts are Table EV1 and Table EV2.

It seems that Table S1 could be part of the methods as Table 1.

- During our routine figure analysis we noticed that the JAK2 staining shown in Figure 3G appears the same as that shown in Supplementary Figure S5B. Please check these panels. If the duplicate use of these images is intentional, please clearly state

that these images are the same in the respective figure legends.

- Our production/data editors have asked you to clarify several points in the figure legends (see below). Please incorporate these changes in the manuscript and return the revised file with tracked changes with your final manuscript submission.

A) Please note that a separate 'Data Information' section is required in the legends of figures 1a-c', e-q; 2a-f; 3e-f, k, n-o; 4d-e, supplementary figures 4g-h; 5b-g; 6a-d; 7e-f; 8d-e.

[Data Information section means that if you define parameters like scale bars or error bars that apply to several panels then please add "Data Information: xyz description..."]

B) Please note that the legend for supplementary figures 2c-e is incorrectly labelled as 2c-d. This needs to be rectified.

C) Please note that the legends for supplementary figures 5b-g are not provided in a sequential manner (legends for figures 5d, f are provided before the legends for figures 5c, e). This needs to be rectified.

D) Although 'n' is provided, please describe the nature of entity for 'n' in the legends of figures 1o, q.

E) Please note that the measure of center for the error bars needs to be defined in the legends of figures 1o, q; 2d, f; 3e-f, k, n-o; 4d-e, supplementary figures 3e; 4g-h; 6d; 7e-f; 8d-e.

- The abstract should be written in present tense and not exceed 175 words. I suggest a shortened version below my signature, but it still has 190 words. Could you please focus and shorten it further?

- Finally, EMBO Reports papers are accompanied online by A) a short (1-2 sentences) summary of the findings and their significance, B) 2-3 bullet points highlighting key results and C) a synopsis image that is 550x300-600 pixels large (width x height) in PNG for JPG format. You can either show a model or key data in the synopsis image. Please note that the size is rather small and that text needs to be readable at the final size. Please send us this information along with the revised manuscript.

- On a different note, I would like to alert you that EMBO Press offers a new format for a video-synopsis of work published with us, which essentially is a short, author-generated film explaining the core findings in hand drawings, and, as we believe, can be very useful to increase visibility of the work. This has proven to offer a nice opportunity for exposure i.p. for the first author(s) of the study. Please see the following link for representative examples and their integration into the article web page:

<https://www.embopress.org/doi/full/10.15252/embj.2019103932>

Kind regards,

Martina

Referee #2:

In this revised manuscript, the authors addressed most of my concerns: it is a shame that they couldn't address the direct link between lobe formation and transcriptional activation due to technical difficulty.

A minor comment concerns Fig 1Q, which is supposed to be a graph showing the impact of CytoB on segregation of Pem1 RNA while its Y-axis is labelled as "lobe scission at st 16".

Abstract suggestion (190 words)

Transcription control is a major determinant of cell fate decisions in somatic tissues. By contrast, early germline fate specification in numerous vertebrate and invertebrate species relies extensively on RNA-level regulation, exerted on asymmetrically inherited maternal supplies. At a later step, maternal-to-zygotic transition is required to complete the deployment of pre-gametic programs in the germline. Here, we focus on early germline specification in the tunicate *Ciona* to study zygotic genome activation. We first demonstrate that a peculiar cellular remodeling event excludes localized postplasmic mRNAs, including Pem-1, which encodes the general inhibitor of transcription. Subsequently, zygotic transcription begins in Pem-1-negative primordial germ cells (PGCs),

as revealed by histochemical detection of elongating RNA Polymerase II (RNAPII), and nascent transcripts from the Mef2 locus. Using PGC-specific Mef2 transcription as a read-out, we uncover a provisional antagonism between JAK and MEK/BMPRI/GSK3 signaling, which controls the onset of zygotic gene expression, following cellular remodeling of PGCs. We propose a 2-step model for the onset of zygotic transcription in the *Ciona* germline, which relies on successive cellular remodeling and JAK inhibition, and discuss the significance of germ plasm dislocation and remodeling in the context of developmental fate specification.

All editorial and formatting issues were resolved by the authors.

Dr. Naoyuki Ohta
Michael Sars for Marine Molecular Biology
University of Bergen
Thormøhlensgt.
55
Bergen, Bergen 5008
Norway

Dear Naoyuki,

I am very pleased to accept your manuscript for publication in the next available issue of EMBO reports. Thank you for your contribution to our journal.

Best regards,
Martina
